# Cellular Prion Protein (PrPc): Putative Interacting Partners and Consequences of the Interaction

**DOI:** 10.3390/ijms21197058

**Published:** 2020-09-25

**Authors:** Hajar Miranzadeh Mahabadi, Changiz Taghibiglou

**Affiliations:** Department of Anatomy, Physiology, Pharmacology, College of Medicine, University of Saskatchewan, Saskatoon, SK S7N 5E5, Canada; h.miranzadeh@usask.ca

**Keywords:** cellular prion protein, protein–protein interactions, prion diseases, gene mutations, signal transduction

## Abstract

Cellular prion protein (PrPc) is a small glycosylphosphatidylinositol (GPI) anchored protein most abundantly found in the outer leaflet of the plasma membrane (PM) in the central nervous system (CNS). PrPc misfolding causes neurodegenerative prion diseases in the CNS. PrPc interacts with a wide range of protein partners because of the intrinsically disordered nature of the protein’s N-terminus. Numerous studies have attempted to decipher the physiological role of the prion protein by searching for proteins which interact with PrPc. Biochemical characteristics and biological functions both appear to be affected by interacting protein partners. The key challenge in identifying a potential interacting partner is to demonstrate that binding to a specific ligand is necessary for cellular physiological function or malfunction. In this review, we have summarized the intracellular and extracellular interacting partners of PrPc and potential consequences of their binding. We also briefly describe prion disease-related mutations at the end of this review.

## 1. Introduction

Cellular prion protein (PrPc) is a glycosylphosphatidylinositol (GPI)-attached, lipid raft-localized protein that is expressed in a variety of cell types, with the greatest abundance in the central nervous system (CNS) [1]. PrPc is encoded by the *PRNP* gene on chromosomes 20 and 2 in humans and mice, respectively. The biosynthesis pathway of PrPc, similar to other membrane and secreted proteins, starts from mRNA translation on the endoplasmic reticulum (ER)-attached ribosomes. After translocation into the ER, the N-terminal signal peptide and the C-terminal GPI anchor addition signal, are released, and the GPI anchor is added to the C-terminal [2]. Translated PrPc peptide transits to the Golgi and is finally delivered to the plasma membrane (PM) [2]. The 208 residue mature form of PrPc contains five octapeptide repeats in the N-terminus that represent the primary Cu^2+^ binding site [3], and a conserved hydrophobic domain in the middle of PrPc [3]. PrPc can be glycosylated at two highly conserved amino acid positions (N180 and N196 in mice) and form diglycosylated, monoglycosylated at N180, monoglycosylated at N196, and unglycosylated PrPc [4]. PrPc adopts a final structure with a flexible N-terminus (amino acids 23–121) and a C-terminal globular domain (amino acids 127–231) containing three α-helices and two β-strands [5].

The PrPc conformational conversion from α-helix-rich to a β-strand-containing protein is followed by the accumulation of amyloid fibrils in the CNS and neurodegeneration of transmissible spongiform encephalopathies (TSEs) [6]. TSEs, also known as prion diseases, belong to a group of progressive degenerative brain disorders including Creutzfeldt–Jakob disease (CJD), Gerstmann–Stra¨ussler–Scheinker disease (GSS), fatal familial insomnia (FFI), and kuru in humans; and bovine spongiform encephalopathy (BSE), chronic wasting disease (CWD), and scrapie in animals [7,8,9].

Although there have been decades of research, a full understanding of the physiological function of PrPc is lacking. Recently, some functions have been attributed to PrPc in the nervous system including neurite outgrowth [10,11], anti-apoptotic roles (during oxidative stress-induced cell death), pro-apoptotic roles (in ER stress) [12,13], Cu^2+^ binding [14,15], synaptic plasticity [16], learning and memory [17], as well as sleep patterns [18]. The best strategy to better elucidate the physiological functions of PrPc is to identify and characterize its interacting protein partners because these interactions are most likely to be parts of physiological pathways in which PrPc plays a role (Figure 1).

Similar to other GPI-anchored proteins, PrPc attaches to lipid raft membrane domains enriched in cholesterol and sphingolipids [19]. Through interactions with PM or extracellular proteins such as laminin, laminin receptors (LR), vitronectin, and caveolin-1, PrPc participates in signal transduction events [20,21]. PrPc may also interact with various intracellular proteins, the majority of which are found in the cytosol, mitochondria, and nucleus [22]. PrPc can also be found in extracellular vesicles such as exosomes, highlighting the role of PrPc in intercellular communication [23]. PrPc and scrapie isoform of prion protein (PrPsc) can both be efficiently transported with extracellular vesicles. The transportation element muskelin in association with cytoplasmic dynein and KIF5C binds to PrPc to efficiently traffic PrPc vesicles. Notably, muskelin handles two-way transport of PrPc and promotes lysosomal degradation over exosomal PrPc release. Muskelin affects neurodegenerative conditions by increasing lysosomal clearance of PrPc, limiting their PM or exosomal presentation [24]. Lack of muskelin induces PrPc recycling to the PM and storage for extracellular trafficking via exosomal carriers. The connection between neuronal intracellular lysosome targeting of PrPc and extracellular exosome trafficking is linked to the pathogenesis of neurodegenerative conditions.

## 2. Transmembrane Forms of Cellular Prion Protein (PrPc)

The majority of PrPc molecules are extracellularly bound to the outer leaflet of the PM via GPI anchors [25]. Moreover, there are also two transmembrane forms of PrPc, ^Ntm^PrPc and ^Ctm^PrPc, being produced in a small amount as part of PrPc normal biosynthesis (Figure 2). These two topologic isoforms are single-spanning membrane proteins with the same transmembrane domain and same hydrophobic region but different orientation in the ER membrane. ^Ntm^PrPc has the N-terminal end in the ER lumen, whereas ^Ctm^PrPc is integrated into the reverse direction displaying the N-terminus on the cytosolic side [26].

## 3. Itracellular Partners of PrPc

Proteins rarely act alone; they interact with other molecules to fulfil their functions instead. Protein–protein interactions are vital in biological systems. The presence of PrPc in different subcellular compartments depends on certain physiological conditions. PrPc overexpression or some PrPc mutations may trigger its retention in the ER and Golgi apparatus. PrPc can be transiently found in the ER and Golgi as it undergoes posttranslational modifications [27]. PrPc is also trafficked to mitochondria during apoptosis and cellular stress. Thus, studies on mitochondria-localized PrPc interaction with mitochondria proteins could help to clarify the mitochondrial dysfunction found during prion pathogenesis [28]. When the octapeptide-containing N-terminus of PrPc is bound to the inner mitochondrial membrane, it can regulate Cu^2+^ trafficking into and out of the mitochondrial matrix during specific cellular conditions [29].

Cellular prion protein (PrPc) can be intracellularly relocated to mitochondria microdomains during cell apoptosis via ER mitochondria-associated membrane (MAM) and the microtubular network, which mediates loss of mitochondrial membrane potential and release of cytochrome C following a contained Ca^2+^ concentration increase [30]. Cellular prion protein (PrPc) relocalization to mitochondrial rafts is crucial for apoptosis execution and cytoskeleton integrity and PrPc has direct or indirect interactions with different cytoskeletal components, including actin, α-actinin, and tubulin [31]. The indirect interaction of PrPc and β-actin is also involved in receptor stability in the PM. Studies on aged mice have revealed an increase in mitochondrial PrPc concomitant with reduced manganese superoxide dismutase activity, release of cytochrome C, caspase-3 induction, and DNA fragmentation [30]. PrPc can play different roles in the cell death response, including intracellular PrPc interaction with BCL2 that prevents the conformational change of BAX [12]. This is, however, an incomplete picture of PrPc biology. To have a clear picture of intracellular PrPc function, the intracellular partners of PrPc should be fully identified and characterized.

A variety of biochemical methods, including immunoprecipitation (IP), yeast two-hybrid (Y2H) method tagged purifications, or affinity purification coupled with mass spectrometry, have been utilized for finding different intracellular PrPc partners (Table 1).

### 3.1. Neuroglobin

Neuroglobin (Ngb), is a globular heme-containing protein with preferential expression in the cytosolic compartment of neurons and retinal cells in all vertebrates [74]. Ngb is involved in neuronal protection during hypoxia, oxidative injury, and cerebral ischemia. Ngb is co-immunoprecipitated (Co-IP) with raft-associated membrane protein flotillin-1 and participates in regulating cell death. Ngb is also a scavenger and antioxidant in Alzheimer’s disease (AD) [75]. The interaction between PrPc and Ngb was discovered by using IP in the retinal ganglion cell layer, and light scattering monitored co-aggregation [32]. The combined approach of automated docking and solid-phase method studies verified that PrPc interacts directly with Ngb via its N-terminus region K23-P28 [33]. PrPc aggregates rapidly upon interacting with Ngb, without causing any misfolding, suggesting a specific role of Ngb as a scavenger of cytosolic prion proteins [32]. As the N-terminal domain is the same in PrPc and PrPsc, Ngb can interact with both forms. PrPc/Ngb co-localization in the retina, where Ngb is abundant, can interfere with PrPc association with other cytoplasmic proteins. This co-localization accounts for the resistance of retinal ganglion cells to PrPsc inoculation and formation of PrPc cytosolic aggregates without cell death [32,33].

### 3.2. Tubulin

Microtubules are highly dynamic cytoskeletal polymers that consist of a lateral association of α- and β-tubulin heterodimers. Microtubules participate in a large variety of cellular functions including cellular self-organization, acting as structural scaffolds, cellular highways, force generators, and signaling platforms [76]. This adaptation is mediated by numerous groups of microtubule-associated proteins (MAPs) [76].

The direct interaction between PrPc and tubulin was first verified by covalent cross-linking of purified proteins [77]. In subsequent studies, co-localization of the N-terminal region of PrPc with microtubules was further confirmed in neuroblastoma N2a cells [78]. The segments of PrPc spanning residues 23–50 and 51–91 of the N-terminus were involved in this interaction [34,35]. The interaction of PrPc with tubulin became stronger, along with increasing numbers of octapeptide repeats. However, only the correct number of octapeptide repeats was essential in regulating the inhibitive effect of PrPc on microtubule polymerization [34]. PrPc N-terminal residues 23–32 are crucial for inducing tubulin oligomerization in both promoting microtubule assembly and preventing microtubule formation conditions [35]. PrPc has a potential role in modulating microtubule dynamics in neurons as a tubulin-sequestering protein. Microtubule dynamics regulation is necessary for post-mitotic neurons, serving critical roles in axon outgrowth, cell signaling, and cell-cell adhesion, among other roles [79]. The active transport of PrPc relies on the intact microtubular network and depends on the dynein and kinesin families of microtubule-associated molecular motors. The complex of PrPc/tubulin involves retrograde and anterograde transport of PrPc along the microtubular track. PrPc residues 23–33 are involved in dynein-dependent transport, whereas residues 53–91 are involved in kinesin-dependent transport [36]. Moreover, Dong and colleagues also demonstrated an association between tubulin and PrPsc with Co-IP in hamster brain homogenate [34].

### 3.3. Tau

Tau is a MAP that is mainly expressed in neurons, although non-neuronal cells present small but detectable quantities of this protein. Tau is a phosphoprotein, promoting and stabilizing microtubule self-assembly by tubulin. Tau also plays an essential role as an axonal microtubule protein [80]. Hyperphosphorylation of Tau can depress microtubule assembly and disrupt the reassembled microtubule [81]. In AD and other neurodegenerative diseases, Tau becomes hyperphosphorylated, which leads to Tau aggregation in the form of paired helical filaments, disassembly of the microtubular cytoskeleton, and impairment of axonal transport. Eventually, Tau aggregation contributes to neuronal apoptosis and neurodegeneration [82].

The Tau protein is detectable in co-immunoprecipitates with both PrPc and PrPsc. The N-terminal domain (residues 23–91) of PrPc interacts with the Tau protein. The evidence of molecular PrPc–Tau interaction highlights the potential role of Tau in the biological function of PrPc [37,38]. The Tau protein has the same interaction site for both PrPc and tubulin, therefore, the Tau protein reduces the PrPc-induced oligomerization of tubulin [39]. The PrPc–Tau interaction can affect the physiological role of the Tau protein, and hence cell viability [40].

### 3.4. Synapsin-1b

Synapsins belong to a family of neuron-specific phosphoproteins that are associated with the cytosolic side of the synaptic vesicles [83]. They are involved in synaptogenesis and neuronal plasticity, including the regulation of synapse development, modulation of neurotransmitter release, and formation of neuron terminals. Their putative function is to control vesicle release through the cross-linking of synaptic vesicles to each other and actin filaments, thus, providing a mechanism for controlling vesicle release. Both PrPc and synapsin proteins are localized at high concentrations in presynaptic nerve terminals. The co-localization of PrPc with synapsin-1b has been shown by Y2H screen with murine PrPc (23–231) and a murine neuronal cDNA library [41]. Co-IP assays have further confirmed this interaction with recombinant and intrinsic proteins in mammalian cells. Synapsin-1b interacts with both the N-terminus (23–100) and C-terminus (90–231) of PrPc. Co-fractionation of PrPc with synapsin-1b in Golgi fractions of N2a cells has verified that these proteins might interact at intracellular vesicles [41]. The interaction between PrPc and synapsin-1b may affect synapsin-1b function and signal transduction. Studies in neuronal cell culture have demonstrated a possible co-regulation of expression for PrPc and synapsin-1. Since both PrPc and synapsin-1b are involved in cell-to-cell contact mechanisms, their expression increases with growing cell density. It is noteworthy that synaptic loss is one of the first pathological features observed in scrapie-infected mice [42,43].

### 3.5. Growth Factor Receptor-Bound Protein 2

Growth factor receptor-bound protein 2 (Grb2) is a ubiquitously expressed adapter protein that interacts with several receptors and intracellular signaling molecules. This adapter protein is distributed in both cytosolic and nuclear fractions. Grb2 is required for a variety of fundamental cellular functions and identified as the main mediator in Ras-mitogen-activated protein kinase (MAPK) activation induced by activated cell surface receptors [84]. Grb2 signaling participates in actin-based cell motility, cell cycle progression, vasculogenesis, angiogenesis, and oncogenesis [85].

Y2H and Co-IP assays conducted with baby hamster kidney (BHK) cells co-expressing PrPc and HA-tagged Grb2 showed an in vitro PrPc–Grb2 interaction, and immunofluorescence microscopy verified that PrPc and Grb2 are co-localized in *Brucella abortus* [41]. The binding site of Grb2 has been mapped to the N- and C-terminal regions of PrPc [44]. The C-terminal interacting region was further narrowed down to residues 100–109 by nuclear magnetic resonance (NMR), circular dichroism, and fluorescence spectroscopy. This domain of PrPc has a positive charge that can bind electrostatically to the negative surface of Grb2-cSH3 [44]. Two single nucleotide polymorphisms that are related to GSS (P102L and P105L) on purified recombinant PrPc 90–231 can disrupt the Co-IP of PrPc and Grb2 [44]. Grb2 and PrPc both interact with synapsin-1b resulting in the formation of a three-protein complex. The binding to Grb2 and synapsin-1b suggests that PrPc could be involved in their signal transduction pathways [41].

### 3.6. Prion Interactor 1

Pint-1 or prion protein interacting protein (PRNPIP) is a 162 aa protein with, as yet, uncharacterized function that directly interacts with PrPc. In mice, it is strongly expressed in the brain, heart, thyroid, and testis. Mouse Pint-1 was identified in a complex with PrPc along with synapsin-1b and Grb2 [41]. The binding motif to Pint-1 was identified as the 90–231 aa region of PrPc [41].

### 3.7. Glial Fibrillary Acidic Protein

Glial fibrillary acidic protein (GFAP) is a prominent intermediate filament protein in reactive astrocytes playing important roles in astrocyte–neuron interactions including suppression of neuronal proliferation and neurite extension in the mature brain, supporting myelination, development of physical barriers to separate damaged tissue, and regulating the blood-brain barrier [86].

The interaction of GFAP and PrPc/PrPsc was identified through ligand blotting from hamster brain homogenates [45] and verified by Co-IP and immunostaining from normal and scrapie-infected brains and recombinant proteins. The regions within PrPc responsible for this interaction are located at the C-terminus of PrPc (residues 91–230) [46,47]. GFAP knockout mice develop normally and are susceptible to PrPsc [87]. GFAP was found to be expressed more in the brain of scrapie-infected mice as compared with non-infected mice [88,89]. The accumulation of PrPsc and elevation of GFAP in the terminal course of TSEs highlight that the association between PrPc and GFAP may contribute to the pathogenesis of prion diseases. GFAP is necessary for cytoarchitectural changes and distribution of reactive astrocytes in prion diseases, and there is also a GFAP-dependent function of glial filaments in reactive astrocytes [90].

### 3.8. Heterogeneous Nuclear Ribonucleoproteins and Aldolase C

Heterogeneous nuclear ribonucleoproteins (hnRNP) belong to a family of RNA-binding proteins involved in mRNA splicing and mRNA transport from the nucleus to cytoplasm.

hnRNP A2/B1 are produced through alternative splicing of a single-copy gene, and B1 isoform has an insertion of 12 amino acids at its N-terminus [91]. hnRNP A2/B1 are expressed in various cell types, including neurons, and are necessary for facilitating transmission of genetic information from nucleus to synapses [92].

Aldolase is a key metabolic enzyme involved in one of the necessary steps in glycolysis. There are three isozymes of aldolase, i.e., A, B, and C. Aldolase A is expressed ubiquitously, whereas aldolase B is mainly expressed in liver and kidney, and aldolase C is expressed more in brain and smooth muscles [93]. mRNA binding activity of aldolase A and C represents a novel function of these isozymes [94].

On ligand blots, purified recombinant PrPc 23–231 binds to the hnRNP A2/B1 and aldolase C from the cytosolic fraction of brain homogenate obtained from PrPc knockout mice and Co-IP with a cytosolic fraction of human Hodgkin’s lymphoma cells confirmed the interaction between these proteins [48]. These data provided the first evidence of binding PrPc to hnRNP and aldolase C and suggested the potential involvement of PrPc in nucleic acid metabolism.

### 3.9. Nuclear Factor Erythroid 2-Related Factor 2

Nuclear factor erythroid 2-related factor 2 (NRF2), a basic leucine zipper (bZIP) transcription factor, regulates the expression of over 200 genes involved in metabolism, immune response, survival, proliferation, autophagy, proteasomal degradation, DNA repair, and mitochondrial physiology [95]. NRF2 is expressed in a wide range of tissues, however, its basal protein levels are usually kept low during unstressed conditions. In the presence of oxidative stress, NRF2 is released from Keap 1. The build-up of NRF2 in the cytoplasm activates nuclear translocation and triggers the transcription of its target genes [95]. Interaction with NRF2 has been found in a screen of a fusion protein composed of mouse PrPc 1–232 and alkaline phosphatase as a probe [49]. It is equally plausible that PrPc may interact with both cytosolic and nuclear fractions of NRF2. Moreover, there is an NRF2 binding site in the prion protein promoter, and NRF2 decreases PrPc expression through this site [96].

### 3.10. B Cell Lymphoma 2

A prominent component of the apoptotic pathway is a family of proteins commonly known as B cell lymphoma 2 (BCL2). This interacting protein family includes inducers and inhibitors of apoptosis [97]. Anti-death BCL2 proteins bind and inhibit pro-apoptotic BCL2 family proteins from regulating the release of cytochrome C from mitochondria, thus, blocking the activation of the caspase cascade and apoptosis pathway. BCL2 proteins have multiple localizations within cells and membranes, likely dictated by their various affinities for different intracellular membranes and binding partners at each location. In addition to the induction of apoptosis, BCL2 has a crucial role in the normal physiological activity of neurons, Ca^2+^ homeostasis, autophagy, mitochondrial dynamics and energetics of normal healthy cells [98].

A Y2H system has been used to identify amino acids 72–254 of PrPc as the interacting residues with BCL2 [50,51]. PrPc constructs targeted to different cellular compartments have revealed the association and co-aggregation of an internal domain of PrPc (amino acids 115–156) with BCL2 in the cytosol [52]. This cytosolic BCL2/PrPc co-aggregation inhibits BCL2 anti-apoptotic function. Cytosolic misfolded PrPc induces apoptosis in neurons, while BCL2 overexpression alleviates the apoptotic effects of cytosolic misfolded PrPc. Further studies analyzed BCL2–PrPc interaction by Co-IP from transiently transfected neuroblastoma cells with cytosolic PrPc and revealed that BCL2 was not co-precipitated with wildtype PrPc [50,52]. Apoptosis was significantly increased after the expression of TSE-linked mutants of PrPc, cytoQ160Stop, and cytoW145Stop, verifying that the region 115–146 of PrPc is the responsible domain for the toxicity of the cytosolic protein [52]. Consequently, the toxic potential of misfolded cytosolic PrPc may result from the co-aggregation with BCL2 and loss of BCL2 function.

### 3.11. 14-3-3 Proteins

The 14-3-3 proteins were initially identified as a family of abundant acidic brain proteins consisting of seven isoforms (β, γ, ε, η, ζ, σ, and τ/θ) in mammals. Structurally, 14-3-3 proteins form homo-and heterodimers [99]. Through interactions with specific phosphoserine or threonine motifs in target proteins, the dimeric complexes of 14-3-3 proteins participate in the regulation of a wide range of biological processes including cell cycle, transcription, signal transduction, apoptosis, and neuronal development. The highest expressions of 14-3-3 proteins are in the brain and 14-3-3 proteins are detected in the cytoplasmic compartment, intracellular organelles, and PM [100]. The 14-3-3 proteins are released from damaged neurons and are detectable in the cerebrospinal fluid of patients with different neurological disorders including mitochondrial myopathy, encephalopathy, GSS, and multiple sclerosis [101,102].

A direct PrPc–14-3-3 interaction has been demonstrated in human brain extract and purified human recombinant 14-3-3 ζ protein in an overlay assay [54]. Immunostaining results revealed an interaction between 14-3-3 ζ and PrPsc in amyloid plaques of CJD patients [53]. Furthermore, studies on amyloid plaques of GSS patients showed strong immunoreactivity of PrPc and 14-3-3 ε isoform [55]. These findings confirmed a direct 14-3-3 interaction with PrPc or PrPsc [103]. The 14-3-3-binding domain is located on the N-terminus of PrPc, residues 106–126 [54,56]. Immunostaining confirmed the co-localization of 14-3-3 proteins and PrPc in the mitochondria of cultured neuronal progenitor cells, and their co-expression was found most prominently in neurons and reactive astrocytes in the human brain. Association and co-localization of heat shock protein 60 (Hsp60) with both 14-3-3 and PrPc have also been previously reported. These observations suggested the existence of a ternary complex of 14-3-3 proteins, Hsp60, and PrPc in the human CNS under physiological conditions and this complex might be altered in pathologic processes of prion diseases [54].

Homology modeling studies revealed that K110 in PrPc is a critical residue for 14-3-3 protein and PrPc interaction [103]. In the same study, inhibition of 14-3-3 dimer formation by a PrPc peptide [104,105,106,107,108,109,110,111,112,113,114,115,116,117,118,119,120,121,122,123,124] was demonstrated, whereas recombinant PrPc promoted it [103]. These results suggest that the neuroprotective role of PrPc and neurotoxic effect of PrPsc could be partly due to their involvement with the 14-3-3 dimerization regulation.

### 3.12. Neurotrophin Receptor-Interacting MAGE Homolog

Neurotrophin receptor-interacting MAGE homolog (NRAGE) has been identified as a cell-death inducer that interacts with the cytoplasmic domain of the p75 neurotrophin receptor (p75NTR). It favors nerve growth factor (NGF)-dependent apoptosis in sympathetic neuron precursors cells, through a Jun N-terminal kinase (JNK)-dependent mitochondrial pathway [104].

A Y2H screen of a rat brain cDNA library demonstrated that cytosolic PrPc constructs interacted with NRAGE. The PrPc–NRAGE interaction was confirmed by an in vitro binding assay and Co-IP using COS-7 cells co-transfected with NRAGE and PrPc. The C-terminal globular domain of PrPc (residues 122–231) was involved in the co-localization with NRAGE. Proteasome inhibition induced PrPc containing C-terminal signal sequence co-aggregation with NRAGE in the cytosol of co-transfected COS-7 cells. Finally, the PrPc–NRAGE interaction appeared to be related to neuronal viability as shown by the effect on mitochondrial membrane potential (an indicator of the early stages of apoptosis) in N2a cells expressing both proteins [57]. These data represent additional evidence confirming the neurotoxic consequence of PrPc interaction with a cytosolic protein, NRAGE.

### 3.13. Casein Kinase II

Protein kinase CK2 (casein kinase II) is a ubiquitous serine/threonine-specific protein kinase, essential for cell proliferation and survival, cancer, apoptosis, angiogenesis, DNA-damage and repair, the ER-stress response, the regulation of carbohydrate metabolism, circadian rhythm, and nervous system function. CK2 can be found in nearly all tissues and the subcellular compartments, although it is especially abundant in nuclei and the soluble cytosol of neurons [105]. CK2 forms a heterotetramer composed of two catalytic (α and/or α^′^) and two noncatalytic subunits (β) [106].

By probing the binding capacity of truncated forms of bovine PrPc and different subunits of CK2, it has been demonstrated that only the C-terminal fragment of PrPc (105–242) interacted with the catalytic subunits of CK2 (α/α^′^). However, the N-terminal fragment of PrPc (25–116) was found to be involved in stimulating the catalytic activity of CK2, most likely through the poly-basic region [58]. The interaction between cytosolic PrPc and CK2 could regulate CK2 activity, and this activation of CK2 resulted in phosphorylation of kinesin light chain, the release of conventional kinesin from membrane-bounded organelles, and fast axonal transport inhibition [60]. These experiments suggest CK2 as a new therapeutic target for prevention of gradual loss of neuronal connectivity that can be seen in prion diseases. Moreover, it has been demonstrated that bovine PrPc can be phosphorylated by CK2 at S154, and this phosphorylation depended on conformational features [59]. Although the mouse PrPc lacks S154, it still interacts with CK2, suggesting that the PrPc–CK2 interaction is different from substrate binding by the enzyme. All three glycosylated forms of PrPc (diglycosylated, monoglycosylated, and unglycosylated PrPc) are involved in the CK2 interaction, although interaction with the unglycosylated form is stronger than glycosylated PrPc [107].

Further studies have demonstrated a significant decrease in CK2 subunit expression in the brain of scrapie-infected hamsters, mice, and human TSE samples [108]. CK2 is one of the few protein kinases that has been observed on the outer leaflet of PM, where PrPc also resides. These findings suggest that CK2 subunit alterations in the brain are disease-correlative phenomena in TSEs and indicate a possible role of CK2 in prion disease pathogenesis. PrPc is present within caveolae; caveolin, the key constituent of caveolae, is both a target and a partner of CK2 [109]. These findings indicate that CK2 can interact with both extracellular and intracellular PrPc.

### 3.14. Mahogunin Ring Finger 1

Mahogunin ring finger 1 (MGRN1), an E3 ubiquitin ligase of the really interesting new gene (RING) finger family, is widely expressed in various tissues including the brain. MGRN1 regulates endosome-to-lysosome trafficking [110]. MGRN1 null mutants exhibit problems in pigment type switching, induce oxidative stress, and show mitochondrial dysfunction commonly associated with neurodegeneration [111,112,113].

Previously, MGRN1 has been shown to interact with the octarepeat domain of both ^Ctm^PrPc and cyPrPc. This interaction can disrupt the function of MGRN1 and trigger pathological changes similar to those observed when MGRN1 expression is knocked down. Depletion of MGRN1 leads to altered lysosomal morphology. Decreasing the function of MGRN1 in cultured cells increases the expression of the cytosolic form of PrPc or transmembrane ^Ctm^PrPc. Interestingly, overexpression or lack of function of MGRN1 does not influence the progression of TSE in mice inoculated with PrPsc [61,62].

### 3.15. Chaperones

Molecular chaperones belong to a group of proteins that assist in the correct folding of polypeptide chains, and regulate the proper assembly of polypeptides into oligomeric structures [114]. There have been several studies on the molecular chaperones involved in the interaction with PrPc including stress-inducible protein 1 (STI1) [115], heat shock protein 60 (Hsp60) [52,63], αβ-crystallin [116,117], calnexin, calreticulin [52,118], BiP [52,119], grp94 [52,120,121], Hsp40, and Hsp70 [52,122].

Hsp60 is a conserved ATP-dependent chaperone [123]. The Hsps belong to a large family of evolutionarily-conserved chaperones with crucial functions in cell development and survival [124]. The interaction site for Hsp60 was mapped for PrPc residues 180–210, which included a portion of the putative α-helices [63].

### 3.16. Members of the Rab Family of Small GTPases

The Rab proteins are small guanosine triphosphate proteins, modulating vesicular trafficking pathways. They are involved in the movement of the proteins between intracellular compartments and PM [65]. Each Rab protein is important for controlling different trafficking steps by promoting vesicle motility, binding, and fusion [125]. Rab7 regulates maturation of early endosomes to late endosomes and modulates transport from late endosomes to lysosomes [126,127]. Activated GTP-bound Rab7 promotes lipid transport, neurite protein degradation, and the trafficking of receptor/ligand complexes [128,129,130]. PrPc cycles between the cell surface and endocytic compartments, and Rab proteins recruit PrPc for trafficking to intracellular compartments [64,65]. There is a potentially direct interaction between PrPc and GTPase family, Rab7a when they are co-localized in the cytosolic area. Silencing of Rab7a by siRNA results in significant changes in PrPc expression and location in the neurons. PrPc accumulation in Rab9-positive endosomal compartments increases when Rab7a is depleted [69]. In addition, prion infection impairs the association of Rab7 with membrane, and may interfere with maturation and lysosomal degradation [131].

### 3.17. Argonaute

Small RNA-guided gene regulation is one of the core principles of cell function that is regulated mainly by members of the argonaute protein family [132]. Argonautes are highly conserved proteins, which interact with microRNAs (miRNAs), short interfering RNAs (siRNAs), and Piwi-interacting RNAs (piRNAs). They are associated with the post-transcription cytoplasmic gene-silencing mechanism [133,134]. Argonautes couple with Dicer to form an RNA-induced silencing complex (RISC)-loading complex (RLC) [135]. To silence transcripts completely or partially complementary to the miRNA, they couple with trinucleotide repeat-containing gene 6 proteins (TNRC6) and form a miRISC complex [136]. Because TNRC6 and Dicer are not able to bind simultaneously to argonautes, endomembranes provide a platform to gather molecular complexes and facilitate temporary interactions between physically separated cellular objects [137]. The interaction between the octarepeat domain of PrPc with argonautes increases the association of argonautes with TNRC6. This is done by removing argonautes from a complex with Dicer and recruiting it to putative regions to form miRISC assembly with TNRC6. Therefore, PrPc is necessary for efficient miRNA-mediated repression [66].

### 3.18. Lactate Dehydrogenase A

Lactate dehydrogenase (LDH), which is widespread in nearly every cell in the cytoplasm, converts lactate to pyruvate [138]. LDH levels differ with each tissue’s metabolic needs, such as growth, biological conditions, and pathological aspects [139,140,141]. The human genome includes the following four LDH genes: LDHA, LDHB, LDBC, and LDHD [142]. Positive effects of lactate on neuronal survival after hypoxia/ischemia are generally known [143]. PrPc and LDHA are known to co-localize. Overactivation of LDHA induces lactate production via PrPc [68]. There is also an interaction between LDHA and Doppel and Shadoo [67], confirming a potentially relevant physiological association of these proteins. Doppel and Shadoo are prion protein family members that are coded by *PRND* and *SPRN,* respectively.

Lactate has a role in the protection of neurons against hypoxia/ischemia, and wild type mice express significantly higher LDH level as compared with PrPc^-/-^. This confirms the potential role of lactate and LDH in mediating the neuroprotective effects of PrPc in hypoxic/ischemic conditions.

### 3.19. Vimentin

Vimentin is an intermediate filament protein involved in providing cell structure and cell movement in several cell types of the CNS [144]. There is a strong interaction between PrPc and vimentin [69,70]. More studies are needed to identify the interactive domain and the physiological relevance of this interaction.

### 3.20. Wnt Signaling Pathway Proteins

The Wnt signaling pathway is an evolutionarily conserved pathway that modulates critical aspects of cell fate determination, cell polarity, cell proliferation, neuronal patterning, and organogenesis. Wnt/β-catenin signaling deficiencies have been identified in many human cancers and different neurodegenerative diseases [145].

In human intestinal cancer cell lines Caco-2/TC7 and SW480 and normal crypt-like HIEC-6 cells, PrPc interacts with the canonical Wnt pathway effectors β-catenin and transcription factor 7-like 2 (TCF7L2) in the cytoplasm and nucleus to upregulate their transcriptional activities [71]. In contrast, the interaction between PrPc and γ-catenin downregulates these transcriptional activities. PrPc also associates with the Hippo pathway effector, Yes-associated protein (YAP), suggesting that PrPc could be involved in gene transcription regulation beyond the β-catenin and TCF7L2 complex [71]. PrPc downregulation and disrupting interaction with Wnt pathway effectors impair Wnt signaling. Therefore, PrPc is a modulator of Wnt signaling.

## 4. The Interaction Partners of PrPc in the Plasma Membrane

The identification of other cellular proteins that PrPc interacts with is crucial for elucidating PrPc’s physiological function in the lipid rafts compartment. Some of these interactors would probably be components of the physiological pathways in which PrPc plays a role. Interactions between PrPc and N-methyl-D-aspartate receptors (NMDARs) reduce overexcitation in neurons [146,147]. PrPc–STI1 interaction induces neuroprotective signals and rescue cells from apoptosis [148,149].

Numerous candidates have been listed as potential binding partners for PrPc in the inner leaflet of PM, intermembrane proteins, or extracellular proteins (Table 2).

### 4.1. Stress-Inducible Phosphoprotein 1

STI1, a conserved co-chaperone, regulates multiple pathways in astrocytes. In addition to its intracellular function as a co-chaperone, STI1 is secreted through extracellular vesicles in a variety of cells [198,199].

Previous studies have confirmed the direct interaction between PrPc and STI1 in the PM. The interaction domain was mapped to amino acids 113–128 of PrPc [149]. Recombinant STI1 can interact with PrPc at the neuronal surface in hippocampal neuron cultures and elicits neuritogenesis in wild type neurons [150]. STI1 binding induces phosphorylation of the MAPK, that is necessary for STI1-promoted neuritogenesis. Furthermore, the interaction between PrPc and STI1 induces Ca^2+^ influx through the α7 nicotinic acetylcholine receptor (α 7nAChR) causing cyclic adenosine monophosphate (cAMP)-dependent protein kinase A activation, and cell death prevention [148,150]. These investigations demonstrated that co-localization of cellular PrPc and its ligand STI1 in hippocampal neurons induced both neuritogenesis and neuroprotection through separate signaling pathways [150]. Further studies have revealed that PrPc engagement with STI1 induced PrPc endocytosis, a necessary step to activate STI1-dependent extracellular signal-regulated kinase (ERK1/2) signaling for neuritogenesis [151]. Additionally, it has been shown that PrPc/STI1 association activated protein synthesis by PI3K/mTOR and ERK1/2 pathways, while protein synthesis stimulation by STI1 in PrPsc-infected cells was impaired [200]. Another study confirmed that the STI1/PrPc association was involved in regulating superoxide dismutase activity (SOD) [201].

### 4.2. Nicotinic Acetylcholine Receptors

There are several classes of nicotinic acetylcholine receptors. There are various homomeric or heteromeric nicotinic receptors in the brain. One of the most abundant nicotinic receptors subtypes in the brain is homomeric α7 nicotinic receptors, which consist of combinations of five α subunits organized in a basic pentameric structure to form a ligand-gated ion channel [202]. The α7nAChR is present in neurons and a wide variety of non-neuronal tissues [203]. The critical roles of α7nAChRs in memory and synapse formation are attributed to their Ca^2+^ permeability [204]. The α7nAchR is activated by pre- and post-synaptic excitation mainly through Ca^2+^ permeability. PrPc expression controls the activation of the α7nAChR, and the interaction between these two proteins is thought to mediate neuroprotective effects in the context of neurodegenerative disorders including prion disease. The physical interaction between PrPc and the α7nAChR was first reported by Beraldo et al. [152]. Moreover, STI1 interaction with PrPc activates Ca^2+^ influx by the α7nAChR and promotes neuronal survival and differentiation.

In an experimental study, a toxic PrPc fragment (aa 106–126) suppressed α7nAChR activation of autophagic flux. Moreover, the α7nAChR activator PNU-282987 induced autophagic flux and protected neurons against PrPc toxic fragment-induced apoptosis. This suggests that α7nAChR-mediated autophagic flux could be involved in the pathogenesis of prion-related diseases [153]. The amyloid β oligomers’ (AβO) clearance is regulated by activation of α7nAchR autophagic flux which improves cognitive function in AD [205,206,207].

### 4.3. Alzheimer’s Disease-Related Proteins

AD is a form of progressive dementia including cognitive impairment, loss of learning, and memory. Interactions between PrPc and other proteins are thought to play an important role in the initiation and progression of AD. AβO accumulation in brain has been recognized as an early toxic event in AD pathogenesis. Amyloid β precursor protein (APP) is a single-pass transmembrane protein with a large extracellular domain that releases AβO into extracellular space by β- and γ-secretases [208]. APP is proposed to function as a cell surface receptor involved in synaptic adhesion and neurotrophic functions [209]. The interaction between PrPc and APP appears to be neuroprotective. The association of PrPc and APP was validated by Co-IP. The PrPc/APP complex is required for CNS development and cell adhesion. Since the PrPc/APP complex plays an important role in synaptic plasticity, the involvement of PrPc in learning and memory, as well as excitotoxicity modulation has also been proposed [154].

Strittmatter and his colleagues found that PrPc functioned as a receptor for misfolded AβO and mediated AβO-induced synaptic dysfunction in COS-7 cells transiently transfected with PrPc cDNA expression plasmids [16], as well as in an Aβ transgenic mice [157]. AβO interacts with the N-terminal domain of PrPc (amino acid 23–27 and 95–110) in AD brains [156,159]. These findings highlight the involvement of PrPc in AD pathogenesis (Figure 3). PrPc has the following two key roles in AD: One, direct interaction of PrPc with β-site amyloid precursor protein cleaving enzyme 1 (BACE1) controls neurotoxic AβO production [155]; and two, PrPc binding to AβO mediates AβO toxicity. In hippocampal slices, the AβO-mediated suppression of long-term potentiation (LTP) depends on PrPc [16], and PrPc is also necessary for memory deficits in AD mouse model [157]. Co-localization of the metabotropic glutamate receptor 5 (mGluR5) and PrPc is critical for the transmission of AβO-mediate neurotoxic signals to intracellular substrates [210]. The mGluRs bind to Gq/11 proteins and stimulate phospholipase C (PLC) activity to make inositol triphosphate (IP3), eventually leading to the Ca^2+^ release from the ER to the cytosol [211]. Since Ca^2+^ is a key second messenger, disruption of Ca^2+^ homeostasis is one of the key causes of neurodegeneration. Moreover, the AβO/PrPc complex activates Fyn kinase, leading to the phosphorylation of the GluN2B subunit of NMDARs [158] and the Tau protein [212]. Low-density lipoprotein receptor-related protein 1 (LRP1) [213] has been identified as co-receptor necessary for the PrPc/AβO complex to activate Fyn kinase and synaptic disruption. LRP1 also mediates eukaryotic translation elongation factor 2 (eEF2) phosphorylation, and dendritic spine loss [210]. In addition, synaptotoxic effects mediated by AβO could be rescued by STI1 through interfering with AβO binding to PrPc or triggering of pro-survival signaling cascades [214].

PrPc associates with mGluR1 and mGluR5 and the interaction domains span residues 51 to 90 and 111 to 134 for mGluR1 and 32 to 114 for mGluR5 [160,162,164,210,215]. Ca^2+^ imaging analysis on N2a cells has revealed that the activation of mGluR1 was modulated by the presence of PrPc and relied on the flexibility of the N-terminal domain of PrPc [162]. PrPc stabilizes the ligand-binding region of mGluR1 to form an open or closed conformation resulting in preservation of the “ready-state” conformation of this domain [161].

The laminin receptor (LRP/LR) is involved in the interaction between AβO and PrPc. LR binds and internalizes PrPc. The LRP/LR has been described as a receptor for different proteins including laminin, PrPc, and its misfolded form PrPsc. Various cell-binding/internalization assays have been used to study LR/LRP’s interaction with PrPc, and the necessity of the laminin receptor for PrPc internalization [165,166]. Mapping analyses of PrPc and LR/LRP interactions have identified PrPc amino acids 144–179 as being involved in direct interaction with LR/LRP, and amino acids 53–93 of PrPc as having an indirect interaction with the heparan sulfate proteoglycan dependent laminin receptor binding site [167]. The indirect interaction of LRP/LR with the octarepeat domain of PrPc/PrPsc in the PM, triggers PrPc internalization through clathrin-coated pits. After prion internalization, the conversion of PrPc into the disease-associated form may take place in endosomes, lysosomes, or endolysosomes [167]. Indeed, the increased LRP/LR level in the brain, spleen, and pancreas of scrapie-infected mice and hamsters confirmed the necessity of LRP/LR for PrPsc propagation and the importance of this interaction in the pathogenesis of prion diseases [165,216]. The proper interactions between basement laminin and the cell membrane are impaired in the absence of PrPc expression, resulting in abnormalities in cell differentiation, especially affecting neuritogenesis and axonal growth [169,170,217].

α-Synuclein is another presynaptic neuronal protein that interacts with PrPc. Abnormal accumulation of aggregated α-synuclein in neurons is the most prevalent clinical feature of the synucleinopathy neurodegenerative diseases. α-Synuclein has been characterized by various formations including monomers, fibrils, and oligomers. Evidence has shown that unfolded cytosolic and membrane-bound forms with α-helical conformation of α-synuclein can be found in the cell. In healthy neurons, the majority of α-synuclein binds to presynaptic membranes [218].

PrPc (residues 93–109) interacts with extracellular α-synuclein oligomers to induce Fyn kinase phosphorylation through mGluR5. The PrPc/α-synuclein complex activates NMDAR and alters Ca^2+^ homeostasis [219].

### 4.4. Laminin

Laminin is a heterotrimeric glycoprotein made up of α, β, and γ chains which are primarily localized to basement membranes. Laminin plays a pivotal role in cell differentiation, proliferation, migration, and death. Laminin is considered to be a highly adhesive molecule that binds to a wide range of extracellular matrix components, including laminin itself and some other molecules on the cell surface [220].

Neuritogenesis, a hallmark of neuronal in vitro differentiation, is strongly dependent on laminin. In particular, laminin’s γ1 C-terminal domain binds to PrPc [169,170]. The γ1 subunit can be found abundantly expressed in the hippocampus, where the γ1 chain has an essential role in axonal regeneration [221]. Laminin binding to PrPc (residues 173–182 of PrPc) modulates neuronal plasticity and memory formation. The signal transduction pathway associated with laminin γ1-PrPc-dependent effects includes the activation of mGluR1/5. This leads to the activation of PLC, Ca^2+^ influx from intracellular stores, and protein kinase C and ERK1/2 activation in primary cultures of neurons. Since laminin also interacts with STI1, it is plausible to suggest a role for PrPc in promoting axon growth in the peripheral nervous system. Indeed, laminin and STI1 engagement to a different domain of PrPc have been shown to act synergistically to transduce neurotrophic signals [171].

### 4.5. GluN2D Subunit of NMDAR

NMDARs are ligand-gated ion channels that are widespread in the CNS and mediate Ca^2+^-permeability requirements of excitatory neurotransmission [222,223,224]. They are involved in key physiological roles, including synaptic plasticity, learning, and memory [225,226]. Consequently, irregular expression levels and changed NMDAR activity have been identified in several neurological disorders and psychiatric conditions [227]. Multiple NMDARs subtypes exist that differ in molecular (subunit) compositions and characteristics. They are assembled as tetrameric channel complexes consisting of two obligatory GluN1 subunits co-assembled with two GluN2 or GluN3 subunits; there are four (GluN2A–GluN2D) and two subtypes (GluN3A and GluN3B) [224]. The GluN2 subunit isoform is a crucial determinant of NMDARs functional properties, such as kinetics of activation, desensitization, and deactivation, with GluN2D subunits exhibiting significantly slower kinetic properties as compared with other GluN2 subunits [228].

PrPc modulates NMDAR in a Cu^2+^-dependent manner [229,230]. PrPc attenuates NMDAR by inducing S-nitrosylation that is supported by PrPc-bound Cu^2+^and NO, produced by NO synthase [230,231]. PrPc-bound Cu^2+^ acts as an electron acceptor, thus allowing NO oxidation and subsequent S-nitrosylation of GluN2A and GluN1 subunits. S-nitrosylation mediates desensitization of NMDAR and prevents overactivation of the NMDAR-coupled ionic channels [229,230,232]. PrPc and GluN2D subunits have been immunoprecipitated from mouse hippocampal homogenate, confirming the GluN2D subunit dependent interaction between NMDAR and PrPc. In PrPc-null mouse neurons, synaptic NMDAR activity was dramatically increased along with an increase in amplitude and slowed kinetics. This was similar to those of GluN2D containing receptors, suggesting enhanced NMDAR activity in the absence of PrPc [146,147]. Recently, Biasini and co-workers reported induction of NMDAR hyperfunction in neurons expressing misfolded PrPc lacking residues 105–125 [233]. Moreover, a recent study revealed that PrPc 90–231 toxicity was mediated by NMDAR-dependent excitotoxicity [234]. Altogether, the data collected from PrPc-null mice and mice lacking normal PrPc function suggest that PrPc may provide neuroprotection by regulating NMDAR activity.

### 4.6. Low-Density Lipoprotein Receptor-Related Protein 1

The low-density lipoprotein receptor family (LRP) consists of different receptors involved in receptor-mediated endocytosis and cellular signaling. This family includes LRP1, LRP2, VLDLR, LRP5, LRP6, LRP1B, and LRAD3. LRP1 is a large membrane protein abundantly expressed in different tissues. LRP1 tends to recognize a wide variety of ligands, both structurally and functionally [235]. LRP1 is a multifunctional cell surface receptor that serves to internalize a large number of extracellular ligands by the clathrin-coated pits [236]. It also regulates signaling cascades in response to different extracellular stimuli, serves as a scaffold receptor to bind ligands, and modulates other membrane proteins [237]. LRP6 is a critical co-receptor for the canonical Wnt pathway, essential for maintaining synaptic integrity and neuronal viability in AD. Wnt binding to the frizzled family of receptors and LRP5/6 induces the intracellular Wnt–β-catenin signaling pathway.

The N-terminal of PrPc co-localizes with the LRP1 both extracellularly and intracellularly inside the biosynthetic compartments [172,173]. More studies are needed to find the exact domain of PrPc involved in the interaction with LRP1. LRP1 downregulation disrupts PrPc endocytosis, confirming its important role in Ca^2+^-mediated endocytosis of PrPc in neurons [172]. Furthermore, sustained LRP1 inhibition by siRNA disrupts PrPc trafficking from biosynthetic compartments to the PM and results in the accumulation of intracellular PrPc molecules [173]. These data reveal that LRP1–PrPc interactions are necessary for both PrPc biosynthesis and endocytic trafficking in neuronal cells. PrPc signaling is related to the negative regulation of glycogen synthase kinase 3β (GSK3β) and potentiates serotonergic signaling by altering the activity of serotonin 1B receptor (5-HT_1B_R), a negative regulator of neurotransmitter release. PrPc and GSK3β coupling are intermediated by Lyn kinase through caveolin-1. The PrPc–LRP1 interaction is required for Lyn kinase to induce PI3K-Akt activation and stimulate downstream inactivation of GSK3β. Collectively, the PrPc/LRP1/Lyn/GSK3β complex supports the release of neurotransmitter and serotonergic signaling [238,239,240].

### 4.7. Neural Cell Adhesion Molecule

The neural cell adhesion molecule (NCAM) is a widely expressed immunoglobulin-like neuronal surface glycoprotein in the periphery and CNS. NCAM mediates a variety of intracellular interactions to regulate adhesion, guidance, and differentiation during neuronal growth [241,242,243]. NCAM is classified based on the three major alternatively spliced isoforms, two 140 kDa and 180 kDa isoforms and a 120 kDa GPI-linked isoform [244]. NCAM 140 and NCAM 180 are localized mostly to non-raft regions of the PM and redistributed to lipid raft regions through palmitoylation of NCAM after NCAM activation.

PrPc has been identified by chemical cross-linking in a complex with NCAM. Peptide array experiments and binding analysis with recombinant PrPc proteins identified that the co-localization occurs between one of the α-helices in the membrane-proximal globular domain of PrPc (amino acids 144–154) and the NCAM [174]. PrPc has direct interactions with NCAM and recruits and stabilizes NCAM in lipid rafts in a heterophilic cis and trans configuration, thus activating Fyn kinase to induce NCAM-dependent neuritogenesis [175]. When these associations are disrupted by PrPc antibodies or in NCAM-deficient and PrPc-deficient neurons, NCAM/PrPc dependent neurite outgrowth is halted, suggesting that PrPc cooperates with NCAM in the nervous system development [175].

### 4.8. Vitronectin

Vitronectin is a multifunctional adhesive glycoprotein present in the extracellular matrix of different tissues in order to regulate cell adhesion, hemostasis, tissue remodeling, tumor metastasis, and immunity [245]. In the CNS, vitronectin is involved in neurite outgrowth induction in neuronal development [246]. Vitronectin has multiple domains for interaction with other protein such as PrPc. The binding site forming PrPc/vitronectin complex is mapped to residues 105–119 of PrPc. The co-localization of PrPc and vitronectin has been shown in mouse embryonic dorsal root ganglia to support axonal growth [21].

### 4.9. Caveolin-1

Caveolins are integral membrane proteins that are the main structural components of caveola membrane invaginations. They are involved in the receptor-independent (caveolin-dependent) endocytosis, and act as scaffolding molecules; caveolins compartmentalize and concentrate various signaling molecules in their vicinities (signal regulation) [247]. The caveolins form a family of three caveolae structural proteins [247]. Caveolin-1 and caveolin-2 are expressed abundantly in almost all tissues except in striated muscle cells, where caveolin-3 is predominant [247]. Caveolae are flask-like cell membrane invaginations; their principle structural protein, caveolin-1, has been shown to regulate localized signaling molecules in caveolae [248]. Caveolin-1, through its caveolin scaffolding domain, binds to various signaling molecules [248]. PrPc and caveolins share common detergent properties and are co-purified in sucrose density gradients. Co-localization of PrPc/PrPsc and caveolin-1 has been confirmed by immunofluorescence and immunoblotting in N2a cells [176,177]. The octarepeat region of PrPc represents the PrPc binding site for caveolin-1 [178]. PrPc is enriched in caveolae structures, with caveolin-1 to activate signal transduction events by Fyn kinase [178,179]. Since PrPc lacks a transmembrane domain, the signaling cascade caused by caveolin-1/PrPc complex is mediated by the PrPc–integrin-β interaction [179].

### 4.10. Integrin β1

Integrins belong to a large family of αβ heterodimeric proteins on the cell surface that link the cytoskeleton’s internal signaling components to the extracellular microenvironment [249]. As essential cell microenvironment sensors, they mediate the transduction of various signals in response to the extracellular matrix structural variations [249]. Signals from the microenvironment are transferred by the integrins with the support of different signaling partners, including adaptor proteins and intracellular protein kinases such as focal adhesion kinase [249,250].

Since PrPc lacks a transmembrane domain, the signaling pathway activated after ligand binding may be attributable to either direct or indirect PrPc interactions with other interacting proteins. The caveolin-1 phosphorylation induced by PrPc stimulation is mediated by integrin β1. There is an interaction between PrPc and integrin β1 [21,179].

PrPc contributes to the formation of neuronal polarization through the regulation of integrin β1 function, fibronectin cell interaction, and cytoskeleton dynamics. PrPc deficiency induces membrane clustering and activation of integrin β1, which facilitates hyperphosphorylation of focal adhesions components, and over activates the RhoA-Rho kinase-LIMK-cofilin signaling pathway, which leads to enhancement of the stability of actin network. It is noteworthy that the negative regulation of PrPc on integrin β1 affects neuritogenesis [217].

### 4.11. Flotillin

Flotillins are ubiquitous highly conserved membrane association proteins, including two homologous proteins, flotillin-1 and flotillin-2 [251]. In mammalian cells, flotillin activates Fyn, MAPK, and Rho-GTPases [252], to affect axon growth and regulation of the cytoskeletal dynamics. Multiple acylations (single palmitate in flotillin-1, a myristate, and three palmitates in flotillin-2), oligomerization, and cholesterol binding enhance the affinity of flotillins for cholesterol-enriched lipid microdomains [253].

PrPc activation by PrPc–PrPc trans interactions between epithelial cells, neurons, and embryonic cells cause PrPc clustering or co-localization with flotillin. This PrPc-flotillin co-localization increases cargo distribution through PrPc-induced and flotillin dependent signaling cascade. PrPc–flotillin interaction seems to regulate trafficking of cargo (exemplified by N-cadherin) to the growth cone to induce neuronal differentiation [180]. In fact, in neuronal cells, PrPc activation can increase PrPc-flotillin complex formation, activate flotillin-associated Fyn and MAPK pathway, direct N-cadherin to flotillin microdomains, growth cone elongation, and neuronal differentiation. PrPc-flotillin complex is also involved in the recruitment of E-cadherin to the cell contact site. Loss of flotillin decreases Src activity and epidermal growth factor receptor (EGFR) phosphorylation, leading to the blockage of EGFR internalization and an elevation of the unphosphorylated form of EGFR in the PM [181]. Further studies are required to characterize the binding regions in PrPc and flotillin.

### 4.12. Epidermal Growth Factor Receptor

The EGFR is a membrane tyrosine kinase receptor for extracellular protein ligands in many types of cells, including the precursors of neurons and glia. EGFR is involved in routine cellular processes, including differentiation, proliferation, and cellular development through this receptor’s binding to various ligands [254]. The EGFR tyrosine kinase domain promotes RAS activation and a subsequent increase in DNA synthesis and cell proliferation [255].

The co-localization of PrPc and EGFR has been detected by Co-IP in N2a cell extracts and human dental pulp-derived stem cells (hDPSCs). PrPc interacts and co-localizes with EGFR in lipid raft fractions and regulates the lipid raft proteins expression and proteins contributing to the vesicular transport and the intracellular trafficking. The PrPc/EGFR complex is also required for the AKT/Cdc42/neuronal Wiskott–Aldrich syndrome protein (N-WASP) pathway to promote neuritogenesis [182,183]. The region of interaction in PrPc is still unclear.

### 4.13. Kainate Receptor GluR6/7 and Postsynaptic Density Protein 95

Kainate receptors are a group of ionotropic glutamate receptors which are widespread in the CNS, where they mediate excitatory synaptic transmission in excitatory neurons. The following five different subunits of the kainite receptors have been identified in two homology groups: KA1, KA2, and GluR5, GluR6 and GluR7 [256]. On the one hand, individual GluR5–7 subunits’ expression, in heterologous systems, results in homomeric receptors that react to glutamate or kainic acid with a rapidly desensitizing current. On the other hand, KA1 and KA2 are active if co-expressed with GluR5, -6, or -7 [257].

Postsynaptic density protein 95 (PSD-95), a membrane-associated guanylate kinase, is the main regulator of synaptic maturation. PSD-95 interacts with, transports, and clusters glutamate receptors to the postsynaptic membrane and links glutamate receptors to different cellular signaling cascades [258,259,260]. PSD-95 also interacts with cytoskeletal linker proteins and cytoplasmic signaling proteins, including Src family protein tyrosine kinase Fyn [261].

There is a large co-localization of PrPc, PSD-95, and GluR6/7 in the postsynaptic density fraction. By forming a protein complex with GluR6/7 and PSD-95, PrPc modulates c-Jun N-terminal kinase 3 (JNK3) pathway activation to inhibit cytotoxic pathways. JNK3 stimulation depends on the interaction of PrPc with PSD-95 and GluR6/7. The susceptibility of PrPc^-/-^ neurons to NMDA, or kainite treatment in hippocampal slices [146,184] is partly attributed to the GluR6/7-PSD-95 complex formation, which triggers JNK3 pathway stimulation and subsequent p53 activation [185,186]. More studies are required to identify the active regions of PrPc in the interaction with PSD-95 and GluR6/7.

### 4.14. Dipeptidyl Peptidase-Like Protein 6

Subthreshold-activating somatodendritic A-type potassium channels have a key role in neuronal signaling and plasticity that relies on their specific cellular position, voltage dependence, and kinetic features [262]. Dipeptidyl peptidase-like protein 6 (DPP6), a protein that is widely expressed in the brain, is an auxiliary subunit of A-type K^+^ channels that directs the functional assembly of K^+^ channels.

PrPc physically associates with DPP6, which results in increased and sustained currents through this channel, attenuating cellular excitability and reduced susceptibility to seizure [187]. The region of interaction of these two proteins is still obscure.

### 4.15. α2δ-1 Subunit of Voltage-Gated Calcium Channels

Voltage-gated Ca^2+^ channels (VGCCs) are heteromeric complexes that are essential mediators for Ca^2+^ entry into the electrically excitable cells [263]. VGCCs are expressed in different tissues, composed of a pore-forming α1 subunit associated with an intracellular β subunit, and an auxiliary α2δ subunit [264]. The α2δ, a membrane-mediated subunit, regulates Ca^2+^ channel current kinetics, controls VGCC trafficking to the PM, and directs the channel to specific presynaptic sites [80,265].

PrPc and α2δ-1 were pulled down from brain extract [188] but the main domain involved in the interaction is still unknown. PrPc null mice studies demonstrated a reduced Ca^2+^ influx through L-type VGCC, suggesting a role for PrPc in VGCC function [266]. Prion-related diseases can be caused by point mutations or insertions in the gene encoding PrPc leading to PrPc misfolding in the ER and its retention in the secretory pathway [189,267,268]. Physical interactions between PrPc and α2δ-1 increase anterograde trafficking and proper synaptic localization and function of the VGCC complex. ER accumulation of misfolded PrPc accumulates α2δ-1 intracellularly, limiting synaptic trafficking of VGCC. Consequently, misfolded PrPc is associated with defective depolarization induced Ca^2+^ influx, irregular cerebellar neurotransmission, and motor dysfunction [188].

### 4.16. Tissue Nonspecific Alkaline Phosphatase

Tissue nonspecific alkaline phosphatase (TNAP) is a member of the alkaline phosphatase family expressed abundantly in different tissues [269]. TNAP is a GPI linked ectoenzyme that hydrolyzes phosphate groups from a wide range of physiological substrates [270,271].

TNAP is an interacting partner of PrPc in lipid microdomains. TNAP and PrPc both interact with extracellular matrix proteins and participate in signaling events [190]. TNAP hydrolyzes phosphate groups of phospho-laminin and supports the interaction of this extracellular matrix protein with PrPc by laminin dephosphorylation. As laminin–PrPc interaction induces neurite outgrowth [170], the neurite retraction happens when the interaction of PrPc/laminin is disrupted by TNAP inhibition. The region of PrPc that associates with TNAP is left to be determined.

### 4.17. Na^+^/K^+^-ATPase

One of the direct PrPc interacting partners is α2/β2-Na^+^/K^+^-ATPase (α2/β2 sodium and potassium ion-exchanging adenosine triphosphatase) [191]. The Na^+^/K^+^-ATPase consists of two different subunits, α and β. The Na^+^/K^+^-ATPase is widely expressed in various cell types where it creates gradients of electrochemical ions that are crucial for many cellular processes [272].

In astrocytes, PrPc functions as a mediator between α2/β2-Na^+^/K^+^-ATPase and the GluA2 subunit of α-amino-3-hydroxy-5-methyl-4-isoxazolepropionic acid receptors (AMPARs) that docks into the adhesion molecule basigin, a protein closely related to lactate transporter monocarboxylate transporter 1 (MCT1) [191]. In the presence of excitatory neurotransmitter glutamate, MCT1 is changed into the active form, resulting in lactate influx or efflux based on the concentration of lactate inside and outside the astrocyte. The functional interplay in regulating astrocyte lactate transport among PrPc, GluA2, α2/β2-Na^+^/K^+^-ATPase, basigin, and MCT1 appears to be prominent in metabolic cross-communication between astrocytes and neurons [191]. The interacting domain of PrPc with Na^+^/K^+^-ATPase remains unclear.

### 4.18. Glypican-1

Glypican-1 (GPC-1) is a heparan sulfate proteoglycan that is a cell-surface and secreted protein that interacts directly with various partners in different cellular pathways [273]. Of the six glypicans in mammals, glypican-1 and -2 are major glypicans involved in brain development [274].

Glypican-1 has GPI anchor co-localizes in lipid rafts compartment with both PrPc and PrPsc. Glypican-1 has been identified as a scaffold for both PrPc and PrPsc to bring them near enough in lipid raft to support prion conversion [192], therefore, glypican-1 is essential for scrapie infection [275,276]. Cell glypican-1 depletion substantially decreases the prion raft association and increases its endocytosis [192]. More studies are needed to clarify the interaction domain of PrPc with GPC-1.

### 4.19. Gpr126 (Adgrg6)

Gpr126/Adgrg6 is a member of the adhesion G protein-coupled receptor (aGPCR) class that plays an essential role in a variety of tissues and organs. Adgrg6 is essential for Schwann cell myelination with major contributions to nerve crush injury repair [277]. PrPc neuronal expression is necessary for myelin physiology [193]. The N-terminal polybasic cluster of PrPc binds in trans to Adgrg6 on Schwann cells to elicit activation of Adgrg6 signals through adenylyl cyclase and elevation of cellular level of cAMP. This stimulates a well-defined downstream signaling cascade that activates transcription factor Egr2 and facilitates the maintenance of myelin [194]. Gpr126^−/−^ mice exhibit a myelin maintenance defect in peripheral nerve Schwann cells that match phenotypes exhibited by PrPc^−/−^ mice [278].

### 4.20. Neuronal Nitric Oxide Synthase (nNOS) and Dystroglycan

Neuronal nitric oxide synthase (nNOS) is an enzyme expressed in the central and peripheral nervous system that is involved in synaptic plasticity in neurons, regulating blood pressure, smooth muscle relaxation, and vasodilation through peripheral nitrergic nerves. NOS catalyzes the five-electron oxidation of l-arginine to generate l-citrulline and the neurotransmitter NO [231]. The majority of nNOS activity is on the inner side of PM. nNOS interacts with raft-associated components of the dystrophin-glycoprotein complex (DGC) [279]. Dystroglycan, the transmembrane protein, is one of the components of DGC that connects the extracellular PrPc to intracellular nNOS in the PM. The interaction between PrPc and nNOS suggests that these two proteins are part of the same functional complex. PrPc can work as a Cu^2+^ chelator and s the complex from the toxic effects of high Cu^2+^ concentrations. Cu^2+^ has an inhibitory effect on the NOS activity. In PrPc knockout mice and PrPsc infected mice, the subcellular localization and function of nNOS are impaired [195]. The putative functional interaction of PrPc with dystroglycan remains to be elucidated.

### 4.21. Potassium Channel Tetramerization Domain Containing 1

Potassium channel tetramerization domain containing 1 (KCTD1) includes BTB/POZ domain (broad-complex, tramtrack, and bric-a-brac), also known as a POZ (poxvirus and Zn^2+^ finger) protein–protein interaction domain, and participates in several potential interactions with itself or other proteins. Deletion mapping studies reveal that PrPc interacts with KCTD1 via the unstructured PrPc residues 51–136 and octapeptide repeats [197]. Since the unstructured region of PrPc is involved in prion aggregation and neurotoxicity, further investigation is needed to clarify the role of PrPc–KCTD1 interaction in the prion aggregation and propagation that can provide new insight to prevent prion diseases.

## 5. PrPc in the Immune System

Since prion diseases are neurodegenerative disorders, much of PrPc research has focused on its function in the nervous system. In addition to the CNS, PrPc is highly expressed in the immune system, in haematopoietic stem cells and myeloid compartments, in human T and B lymphocytes, natural killer (NK) cells, platelets, monocytes, dendritic cells, and follicular dendritic cells. Although the prion protein is abundantly expressed in the immune system, it has been reported that PrPc^−/−^ mice have only minor abnormalities in immune function. Increased expression of surface PrPc has been documented during T-cell activation. In human T cells, PrPc co-localizes with F-actin, CD3, Fyn, reggie-1/2, lymphocyte-specific tyrosine kinase P56lck (Lck), zeta chain associated protein 70 kDa (Zap70), linker for activation of T cells (LAT), NADPH, and MAPKs shortly after T cell polyclonal activation [280]. These interactions increase the release of reactive oxygen species (ROS) and increase Src family kinases activation and ERK1/2 phosphorylation [280].

The other immune molecule that interacts with PrPc is tetraspanin-7. Tetraspanin-7 is a member of the tetraspanin family that is comprised of membrane-spanning proteins with a conserved structure, and mainly acts as the organizer of membrane proteins [281]. Tetraspanin-7 is widely expressed in the CNS including the cerebral cortex and hippocampus. Using Co-IP, Y2H, and confocal microscopy, tetraspanin-7 has been identified as a potential interacting partner of PrPc in HeLa cells. PrPc interacts with tetraspanin-7 via the peptide region 154–182 of bovine PrPc [282]. The PrPc association with immune molecule tetraspanin-7 may help to clarify the response of immune cells to PrPc. Tetraspanin-7 is linked to some neuropsychiatric diseases such as Huntington’s chorea and X-linked mental retardation [283]. The PrPc association with tetraspanin-7 supports the idea that this interaction may play a role in altered immune responses in prion diseases.

## 6. Polymorphisms and Mutations on the *PRNP* Gene

Different pathogenic mutations have been identified in the coding region of the *PRNP* gene. Mutations may change a single residue in PrPc, insert or delete amino acids, or cause an unusual short version of PrPc. Despite many studies on prion diseases, the role of pathogenic mutations on the generation of a misfolded PrPc and how pathogenic *PRNP* mutations are involved in prion diseases has yet to be resolved. The main factor in the pathogenesis of infectious, inherited, and sporadic prion diseases is the misfolding of the normal form of PrPc into the protease-resistant-β sheet-rich isoform, PrPsc. In the inherited forms of prion diseases, genetic mutations in the *PRNP* gene may induce conformational changes, but the process of conformational alteration has not been completely understood. The mature form of PrPc is composed of flexible N-terminal domain and C-terminal globular region [284].

Most of the identified pathogenic mutations are in the C-terminal domain [285], in the β2-α2 loop region (residues 165–175), and the α2-α3 inter-helical interfaces (residues 185–200) of PrPc. These two regions are involved in the flexibility variation of PrPc protein [286]. The flexibility within the variation promotes the access to distinct protein conformational state, remodeling the region for molecular recognition events such as protein–protein and protein–ligand interactions [287].

The N-terminal domain is associated with fibrillation and the determination of the physical characteristics of disease-related PrPc isoforms. Most of the functional advantages of intrinsically disordered regions are due to the lack of intact tertiary and secondary structures. These advantages are described by the disorder to order change, promoting binding rate flexibility of the interaction with various partners and specific low-affinity binding [284]. Moreover, these characteristics of disordered, unstructured proteins are necessary for post-translational modifications that are involved in low affinity and high-specificity interactions between a protein and its partners. The N-terminal domain of PrPc is highly conserved among species. The capability of PrPc to interact with different intra- and extracellular partners relies on its unstructured N-terminal domain with specific conserved and non-random amino acid sequences [284]. There are also some mutations in the N-terminal domain of PrPc. The N-terminal polybasic region (NH2-KKRPKP) is associated with PrPc internalization, regulates correct folding of PrPc, and also mediates the acquisition of strain-specific structure in the disease [288]. Furthermore, residues 23–50 play a protective role against intracellular ROS levels [289]. The N-terminal domain in PrPc is very important for its physiological functions as a mutation in this domain increases the susceptibility to prion diseases.

*PRNP* mutations that induce conformational changes in PrPc may affect the interaction of PrPc with specific partners. Thus, studies of *PRNP* mutations can provide insights into the specific role of PrPc interactions with ligands and their signaling pathways. There are some pathological mutations related to the specific clinical group of prion diseases. The inherited prion disease diagnosis and subclassification have been done based on these mutations (Figure 4). Inherited human prion diseases include familial CJD, GSS, and fatal familial insomnia (FFI). Other mutations are related to a range of clinical and pathological phenotypes that are different across and within families with the same mutation [285], with very striking phenotypic heterogeneity. Furthermore, various mutations of the *PRNP* gene have been identified to play an important role in clinical pictures imitating other neurodegenerative diseases including cerebral amyloid angiopathy (CAA) [290], frontotemporal dementia (FTD) [291,292,293,294,295], familial neuropsychiatric illness [296], Huntington’s disease [297], and familial AD [298].

## 7. Conclusions

PrPc co-localizes with multiple ligands to act on a variety of targets simultaneously. PrPc can act as a cell signaling molecule, functioning as a receptor and scaffold for different ligands and activating a variety of signal transduction pathways. Prion diseases are identified by a conformational transition in the normal host protein PrPc. Although most of the mature PrPcs are attached to the PM through GPI anchor, topologically unnatural variants of this protein can occur during its biosynthesis. There is growing evidence that supports the fact that the mis-localized PrPc can bind to and aggregate cytosolic proteins which are not its physiological partners. This protein aggregation can lead to a loss of function of the protein partners and can provide a molecular mechanism for the cyPrPc neurotoxicity in prion diseases. The cyPrPc, at low concentration, can modulate the physiological functions of the cytosolic protein partners, whereas it can dysregulate them at high, pathological levels causing neurodegeneration.

Authors are aware and acknowledge that some of the techniques used to study interacting partners of PrPc may have limitations and pitfalls similar to any other protein–protein interaction research. Some of these technical shortcomings and pitfalls have been discussed in a review by Hayes et al. [299]. It should also be considered that some proteins in biological samples may interact with the surface of their containers. It is also noteworthy that both physical adsorption (physisorption) and chemical adsorption (chemisorption) phenomena may exist with regards to analytes and, in particular, proteins including PrPc [300]. The nature of PrPc interactions can be weak or short-lived or temporary, which makes their interactions very difficult to study with the limitations in analytical techniques. Nonetheless, the identification and characterization of these interactions are imperative for a better understanding of the physiology and pathophysiological roles of PrPc. PrPc, an evolutionary conserved molecule among the species, appears to need further studies because a clear description of PrPc function still requires more studies [301,302]. Understanding different partners of PrPc and identifying interacting domains may assist us to better understand underlying molecular mechanisms in a variety of neurological and psychiatric diseases, as well as diseases resulted from mutations in PrPc gene and its interacting partners.

## Figures and Tables

**Figure 1 ijms-21-07058-f001:**
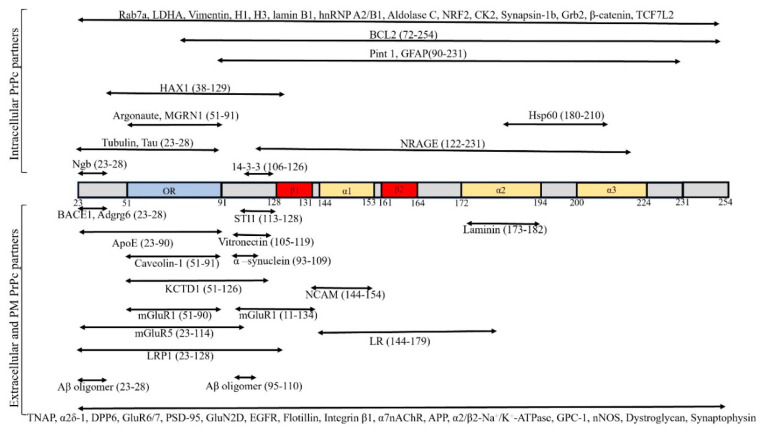
Cellular prion protein (PrPc)-binding partners. The translated form of PrPc is shown with major domains in color (amino acid residue numbers are for mouse PrPc). Each binding protein is indicated together with the stretch of amino acid residues including the binding domain in mouse PrPc. Lactate dehydrogenase A (LDHA), Histone 1/3 (H1/3), Glial fibrillary acidic protein (GFAP), Neurotrophin receptor-interacting MAGE homolog (NRAGE), Neural cell adhesion molecule (NCAM), laminin receptors (LR), Epidermal growth factor receptor (EGFR).

**Figure 2 ijms-21-07058-f002:**
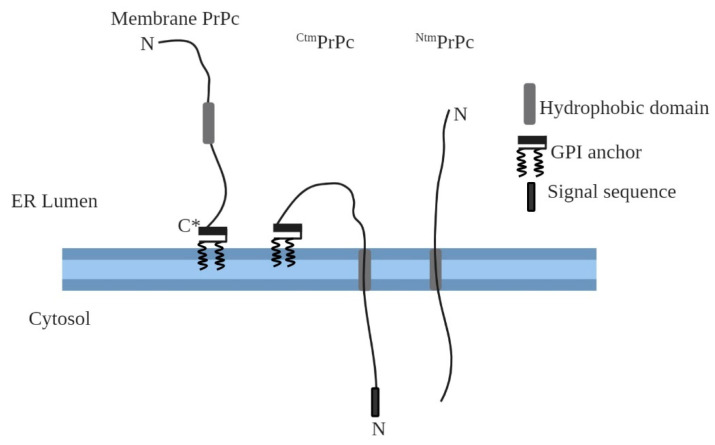
Transmembrane forms of PrPc. The transmembrane forms of PrPc, ^Ntm^PrPc and ^Ctm^PrPc, span the lipid bilayer once through the hydrophobic domain with either C-terminal or N-terminal, respectively on the cytosol side of endoplasmic reticulum (ER). C* represents the correctly positioned PrPc.

**Figure 3 ijms-21-07058-f003:**
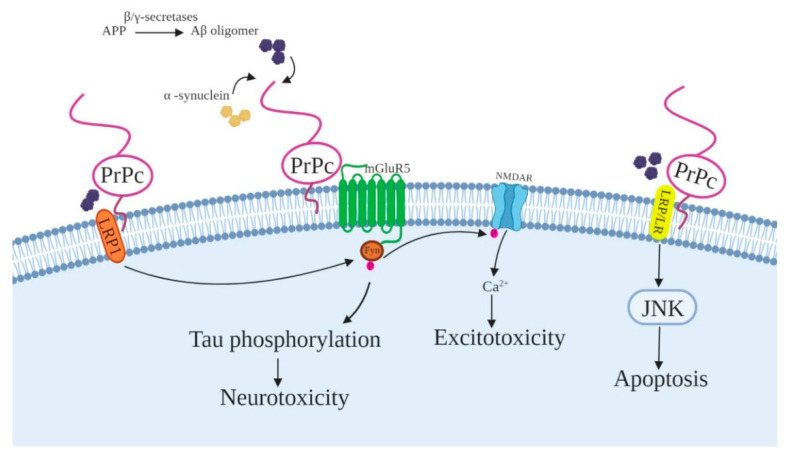
The molecular events of PrPc-ligand interaction in Alzheimer’s disease (AD). PrPc’s interaction with amyloid beta oligomer (AβO), low-density lipoprotein receptor-related protein 1 (LRP1), laminin receptor (LRP/LR), and α-synuclein induce Fyn activation, leading to Ca^2+^ influx through phosphorylated N-methyl-D-aspartate receptors (NMDAR) and dendritic spine destabilization. Moreover, Fyn kinase can induce Tau phosphorylation. LRP1 is also involved in PrPc/AβO-mediated Fyn activation. LRP/LR, a transmembrane receptor, plays an important role in apoptotic signaling pathway by interacting with PrPc and AβO.

**Figure 4 ijms-21-07058-f004:**
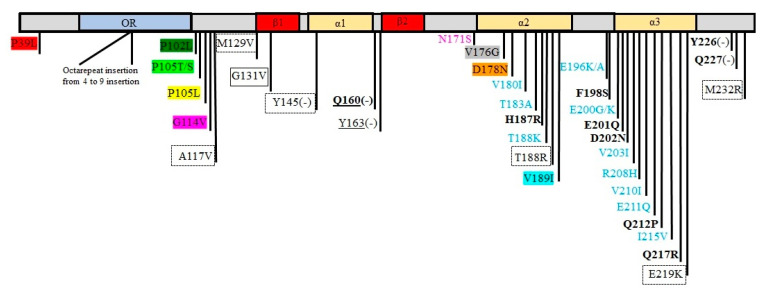
*PRNP* gene single nucleotide polymorphisms. Schematic description of the *PRNP* gene, with the main polymorphisms associated with prion diseases. Mutations with red highlight associated with (frontotemporal lobar degeneration (FTLD) syndrome and frontotemporal dementia (FTD)), octapeptide insertions Creutzfeldt–Jakob disease (CJD), dark green highlight (classical CJD-like symptoms, Gerstmann–Stra¨ussler–Scheinker disease (GSS)), yellow (GSS, spastic paraparesis and progressive dementia), light green highlight (GSS), Pink highlight (CJD, neuropsychiatric symptoms), dotted box (CJD, progressive cortical dementia and cerebellar ataxia), solid box (GSS, tremor and apraxia), underline (Alzheimer’s disease (AD)-type pathology), grey highlight (cerebellum ataxia, personality changes, and progressive dementia), orange highlight (CJD and fatal familial insomnia (FFI) depend on the allele on codon 129, Met, or Val), blue highlight (classical and atypical CJD (behavioral abnormalities, ataxia, and extrapyramidal features)), blue (classical and atypical CJD) and bold (classical and atypical GSS).

**Table 1 ijms-21-07058-t001:** Intracellular interacting partners of PrPc.

PrPc Interacting Protein/Function	Cellular Localization	Effect of Interaction	Technique	References
Neuroglobin (Ngb)/hemoglobin	Cytoplasm	PrPc aggregation	Immunostaining, docking	[32,33]
Tubulin/cytoskeletal protein	Cytoplasm	Inhibit microtubule polymerization, Tubulin oligomerization, Retrograde and anterograde transport of PrPc	Co-IP, pull-down, co-fractionation, crosslinking	[34,35,36]
Tau/microtubule-associated protein	Cytoplasm	Reduction of PrPc induced oligomerization of tubulin	Co-IP, pull-down	[37,38,39,40]
Synapsin-1b/neuron-specific phosphoprotein	Cytoplasm	Unknown	Cofractionation, Y2H, Co-IP	[41,42,43]
Growth factor receptor-bound protein 2 (Grb2)/growth factor receptor	Cytoplasm, Nucleus	Unknown	Cofractionation, Y2H, Co-IP	[41,44]
Pint-1/unknown	Cytoplasm	Unknown	Y2H, Co-IP	[41]
Glial fibrillary acidic protein (GFAP)/intermediate filament	Cytoplasm	Unknown	Co-IP, pull-down, overlay	[45,46,47]
Heterogeneous nuclear ribonucleoproteins (hnRNP) A2/B1/RNA-binding protein	Cytoplasm, Nucleus	Unknown	Co-IP, overlay	[48]
Aldolase C/metabolic enzyme	Cytoplasm	Unknown	Co-IP, overlay	[48]
Nuclear factor erythroid 2-related factor 2 (NRF2)/transcription factor	Cytoplasm, nucleus	Unknown	Screen of bacteriophage expression library of brain cDNA	[49]
B cell lymphoma 2 (BCL2)/apoptosis regulator	Cytoplasm	PrPc aggregation, Inhibits BCL2 anti-apoptotic function	Co-IP, Y2H, affinity	[50,51,52]
14-3-3 protein/phosphorylation-dependent scaffold protein	Cytoplasm	Unknown	Co-IP, pull-down, overlay	[53,54,55,56]
Neurotrophin receptor-interacting MAGE homolog (NRAGE)/cell-death inducer	Cytoplasm	Aggregation, changing mitochondrial membrane potential	Co-IP, pull-down, Y2H	[57]
Casein kinase II (CK2)/serine-threonine kinase	Cytoplasm, nucleus, extracellular matrix	PrPc phosphorylation, regulates CK2 enzymatic activity	Co-IP, pull-down, overlay, surface plasmon resonance	[58,59,60]
Mahogunin ring finger 1 (MGRN1)/ubiquitin ligase	Cytoplasm	Aggregation, disruption of mahogunin function, neurodegeneration	Pull-down, Co-IP, Y2H	[61,62]
Heat shock protein 60 (Hsp60)/chaperon	Mitochondria	Unknown	Y2H	[63]
Members of the Rab family of small GTPases/guanosine triphosphate proteins	Cytoplasm	Intracellular PrPc trafficking	Co-IP	[64,65]
Argonaute/small RNA binding protein	Cytoplasm	Posttranscription cytoplasmic gene-silencing mechanism	Pull-down, electron microscopy	[66]
Lactate dehydrogenase A (LDHA)/oxidoreductases enzyme	Cytoplasm	Activation of LDHA, neuroprotection	Co-IP	[67,68]
Vimentin/cytoskeletal protein	Cytoplasm	Regulate intracellular transportation	Co-IP	[69,70]
β-catenin/transcriptional coactivator	Cytoplasm, nucleus	Transcriptional regulation	Co-IP	[71]
Transcription factor 7-like 2 (TCF7L2)/transcription factor	Cytoplasm, nucleus	Transcriptional regulation	Co-IP	[71]
Histone H1/3, lamin B1/structural chromatin components	Nucleus	Transcriptional regulation	Far-Western blot	[72]
HS-1-associated protein X1 (HAX-1)/apoptosis regulator	Cytoplasm	Oxidative stress, antiapoptosis	Co-IP, microarray	[73]

**Table 2 ijms-21-07058-t002:** Extracellular and plasma membrane partners of PrPc.

PrPc Interacting Protein/Function	Cellular Localization	Effect of Interaction	Technique	References
Stress-inducible phosphoprotein 1 (STI1)/extracellular ligands	PM	Ca^2+^ influx induction, neuritogenesis, neuroprotection	Co-IP, pull-down, complementary hydropathy	[148,149,150,151]
Nicotinic acetylcholine receptor (α7nAChR)/neurotransmitter receptors	PM	Mediate autophagic flux, Ca^2+^ signaling	Co-IP	[152,153]
Amyloid beta precursor protein (APP)/cell adhesion receptors	PM	Cell adhesion, CNS development	Co-IP	[154]
β-site amyloid precursor protein cleaving enzyme 1 (BACE1)/membrane-bound aspartic protease	PM	APP cleavage	Co-IP	[155]
Amyloid beta oligomer (AβO)/ extracellular ligand	PM	Neurotoxicity, LTP suppression, NMDAR phosphorylation, Tau phosphorylation	Co-IP, immunostaining	[16,156,157,158,159]
Metabotropic glutamate receptor 1 (mGluR1)/neurotransmitter receptors	PM	Neurite outgrowth	Co-IP, immunostaining	[160,161,162]
Metabotropic glutamate receptor 5 (mGluR5)/neurotransmitter receptors	PM	AβO -mediate neurotoxicity, promoting long term depression and inhibiting long-term potentiation, neurite outgrowth	Co-IP, immunostaining	[160,163,164]
Laminin receptor (LRP/LR)/ cell adhesion receptors	PM	PrPc internalization, Cell survival	Co-IP, Y2H	[165,166,167]
α-synuclein/neurotransmitter release regulator	PM	Ca^2+^ influx	Co-IP	[168]
Laminin/Extracellular matrix protein	Extracellular matrix	Modulate neuronal plasticity and memory	Pull-down	[169,170,171]
N-methyl-D-aspartate (NMDA) (GluN2D)/neurotransmitter receptors	PM	Neuroprotection	Co-IP	[146,147]
Low-density lipoprotein receptor-related protein 1 (LRP1)/endocytic receptor	PM, Intracellular compartments	Endocytosis and trafficking of PrPc, stimulation of neurotransmitter release	Co-IP	[172,173]
Neural cell adhesion molecule (NCAM)/ cell adhesion receptors	PM	Neuritogenesis, neurons development	Co-IP, immunostaining, pull-down	[174,175]
Vitronectin/extracellular matrix protein	Extracellular matrix	Axonal growth	Immunostaining, pull-down	[21]
Caveolin-1/scaffold proteins	PM	Regulate exosome secretion, regulate Src kinase Fyn activation	Co-IP, immunostaining, pull-down	[176,177,178,179]
Integrin β1/cell adhesion receptors	PM	Neuritogenesis	Co-IP, immunostaining, pull-down, overlay	[21,179]
Flotillin/scaffold proteins	PM	Neurons differentiation by trafficking of N-cadherin, trafficking of EGFR	Co-IP, immunostaining, electron microscopy	[180,181]
Epidermal growth factor receptor (EGFR)/growth factor receptors	PM	Neuritogenesis	Co-IP	[182,183]
Postsynaptic density protein 95 (PSD-95)/scaffolding protein	PM	Protection against excitotoxicity	Co-IP, cDNA library	[146,182,183,184,185,186]
Kainate receptor GluR6/7/ ionotropic glutamate receptors	PM	Protection against excitotoxicity	Co-IP	[146,182,183,184,185,186]
Dipeptidyl peptidase-like protein 6 (DPP6)/potassium channel	PM	Downregulation of neuronal membrane excitability	Co_IP	[187]
α2δ-1 subunit of voltage-gated/calcium channel	PM	Ca^2+^ influx	Co-IP, immunostaining	[188,189]
Tissue nonspecific alkaline phosphatase (TNAP)/ectoenzyme	PM	Laminin dephosphorylation	Co-IP, immunostaining, mass spectrometry	[190]
α2/β2-Na^+^/K^+^-ATPase/neurotransmitter receptors	PM	Lactate transportation in astrocytic	Co-IP, immunoaffinity chromatography, pull-down	[191]
Glypican-1 (GPC-1)/growth factor regulator	PM	Lipid raft localization of PrPc	Co-IP, immunostaining	[192]
G protein-coupled receptor (Adgrg6)/adhesion G protein-coupled receptor	PM	Myelin maintenance in the peripheral nerves	Co-IP, immunostaining, electron microscopy	[193,194]
Neuronal nitric oxide synthase (nNOS)/flavo-hemoprotein	PM	Modulate nNOS activity	Co-IP, immunostaining	[195]
Dystroglycan/integral membrane glycoprotein	PM	Unknown	Co-IP, immunostaining	[195]
Synaptophysin/Integral membrane glycoprotein	Synaptic vesicles	Unknown	Co-IP, immunostaining	[195]
ApoE/cholesterol carrier	Extracellular matrix	Unknown	Co-IP, pull-down	[196]
Potassium channel tetramerization domain containing 1 (KCTD1)/ion channel	PM	Unknown	Co-IP, Y2H	[197]

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
