# Peer review of "Cellular Prion Protein (PrPc): Putative Interacting Partners and Consequences of the Interaction"

_ijms, 2020, doi:10.3390/ijms21197058_

Round 1

Reviewer 1 Report

This review manuscript describes functional associations of cellular prion protein with its interacting partners. In particular, it is focused on the intracellular and extracellular partners of the PrPcprotein and the potential consequences of their binding.

The content appears quit novel and contributes to the knowledge of the functional role of PrPc. However, some points have to be improved before it could be considered for publication:

  • The title should be modified: “its interacting partners” is better. In addition, the functional approach of this review could be also clarified in the title.
  • A possible localization of PrPcin mitochondria and/or MAMs should be considered.
  • Functional interaction of PrPcwith interacting partners involved in T cell activation should be reported in a specific paragraph.
  • Interaction with cytoskeletal proteins during apoptosis should be considered. Furthermore, discussing shortly the role of PrPc and the interacting partners during the apoptotic process will improve this review.
  • Could you describe the possible interaction of PrPcwith other LRP, such as LRP6 (and other components of Wnt pathway), pointing out its possible role in physiological and pathological conditions?
  • Although the review appears to be quite complete, 321 references are too many!

Author Response

Reviewer 1:

The title should be modified: “its interacting partners” is better. In addition, the functional approach of this review could be also clarified in the title.

Thank you for the Reviewer 1 suggestion. We changed the title to “Cellular prion protein (PrPc): putative interacting partners and consequences of the interaction” to better clarify the content of this review.

A possible localization of PrP in mitochondria and/or MAMs should be considered.

We added more details about the intracellular localization of PrPc in chapter 3. (line 90-103, highlighted in yellow)

Functional interaction of PrP with interacting partners involved in T cell activation should be reported in a specific paragraph.

Although the main focus of this manuscript is the interacting partners of PrPc in the nervous system, to follow the respected reviewer’s suggestion, we just added a section (“PrPc in the immune system”) about the involvement of PrPc in the immune system and the PrPc partners in T cell activation in chapter 5.(line 867-878, highlighted in green)

Interaction with cytoskeletal proteins during apoptosis should be considered. Furthermore, discussing shortly the role of PrP and the interacting partners during the apoptotic process will improve this review.

The interaction between PrPc and cytoskeletal proteins, including actin, α-actinin and tubulin plays role in the re-localization of PrPc to mitochondria. we therefore added more details to chapter 3, intracellular partners of PrPc. (line 103-106, highlighted in gray)

Both intracellular and plasma membrane partners of PrPc play role in the apoptosis process. To follow the reviewer 1 suggestion, we added more details about the apoptosis process in both chapter 3, “intracellular partners of PrPc”(line 103-110, highlighted in gray) and chapter 4, “The interaction partners of PrPc in the PM”.(line 439-441, highlighted in turquoise)

Could you describe the possible interaction of PrP with other LRP, such as LRP6 (and other components of Wnt pathway), pointing out its possible role in physiological and pathological conditions?

Based on our literature search, PrPc does not interact with LRP6 but toxic effects of neurotoxic prion peptide (PrP106-126) is being alleviated by the repressor element 1-silencing transcription factor (REST) through the activation of LRP6-Wnt-β-catenin signaling pathways (1- Song Z, et al. Oncotarget. 2016 Mar 15;7(11):12035-52. doi: 10.18632/oncotarget.7640.; 2- Song Z, et al. Front Mol Neurosci. 2017 May 3;10:128. doi: 10.3389/fnmol.2017.00128. eCollection 2017.). In agreement with the Reviewer 1 comment, We added the PrPc interactions with Wnt pathway effectors, β-catenin and transcription factor 7–like 2 (TCF7L2).

Please refer to the chapter (line 3-20). Wnt signaling pathway proteins. (line 421-433, highlighted in green)

Although the review appears to be quite complete, 321 references are too many!

We reduced the number of references to 312 references.

Reviewer 2 Report

The authors of the current manuscript entitled “Cellular prion protein and it`s interacting partners” (submitted as a Review Article) undertook the impressive and surely laborious effort to summarize proteins described to bind to or even functionally interact with the cellular prion protein. This task deserves all due respect, and a comprehensive article in this regard will likely be a valuable resource and overview for researchers in the field and prospective interested readers from other areas as well.  

However, in my opinion there are several weaknesses that should be carefully and consequently addressed by the authors to significantly improve the overall quality prior to acceptance for publication:

A)

The whole manuscript is filled with errors regarding grammatical issues, orthography, syntax and punctuation. Unfortunately, this makes it very difficult and tedious for readers to follow the overall flow of the text. For instance, many sentences are either lacking verbs or have it in the wrong place or form (e.g. tense, or plural vs singular); the article “the” is often used unnecessarily while missing elsewhere; “N-terminal” or “C-terminal” are adjectives, not nouns. Therefore, the text should undergo a careful and thorough proofreading by native English speakers. Also, several formatting errors can be found throughout the manuscript.

B)

For being a review on this topic, a more critical appraisal (maybe even deserving a separate chapter) would be highly appreciated: Can really all described “interactions” be considered “true” or -even more importantly- “physiologically relevant”? What about potential artefacts? A pull-down (e.g. from brain homogenates) may reveal several false “interactions”. What about technical limitations? PrP is known to be a “sticky” protein (note the use of detergents [e.g. 0.03% CHAPS] in recent ELISA studies of CSF samples to reduce PrP binding to plastic surfaces [group of Vallabh/Minikel]), so much “binding” to other proteins might be seen in lysates/homogenates without any in vivo relevance. Also, some parts/referenced studies deal with the so-called “neurotoxic” prion peptide 106-126 – is this really a good proxy/analog to PrPC? I wonder how the multitude of interactions can really be true (considering protein/membrane topology and other constraints), yet I did not have the impression that the authors are wondering about this, too. Readers would likely be thankful for some “guidance” to understand these issues. What is the authors` or the referenced studies` view/explanation on this? How can PrPC interact with cytosolic proteins? Or is this interaction only suggested for cytosolic PrP or the other altered topology versions (if so, this should be clarified better)? The respective chapter 2 is somewhat confusing y the way – a scheme would surely be helpful (see point C below). Also, the individual subchapters introducing the particular proteins are mostly filled by a row of sentences mentioning -partially loosely connected- findings/facts. A bit more of conceptual guidance would clearly improve understanding.

C)

Figures/schemes: It is somewhat strange that the only figure in a comprehensive review on PrP interactors is dealing with PrP mutations. Authors should consider additional (and more relevant) pictures/schemes (e.g. to improve understanding of topologically altered PrP forms, or to illustrate some important (and well-accepted) interactions and their (patho)physiological outcome).

D)

Chapter 5 (Mutants/Polymorphisms) is not really connected well to the major topic of “interactions” – authors should elaborate on this a bit further to better integrate this aspect.

E)

Several statements are misleading, confusing or even wrong. For instance, not whole PrPC -only its N-terminal half- is intrinsically disordered (Abstract); “purified from” is often used although “expression” is meant; line 184ff: “GFAP was found to be expressed more in the brain of scrapie-infected when compared to wild animals…” – I guess that “wild-type” is meant here (yet still, the comparison likely was between infected and non-infected mice, right?); various spellings can sometimes be found (e.g. Pint1 vs Pint 1 vs pint-1); line 244ff: “The highest tissue concentration of … is the brain and detects in …”; it is not true that A-beta is not found in non-demented controls; etc.)

F)

The conclusion contains some remarkable sentences that are much too general and/or much too far-reaching/exaggerated: e.g. “PrPC is identified as the main controller of cellular signalling …” or “… have still some undisclosed crucial roles…” (how can something unknown or undiscovered already be important or relevant?)

G)

The tables are surely helpful. However, I suggest to include (i) full names & abbreviation, (ii) information on the suggested main/primary function of that particular protein (class) (e.g. Pint-1 (also termed PRNPIP or ERI3 is described as a RNA exonuclease), (iii) how / by which method and in which system interaction was described.

H)

Assignment: some listed proteins (e.g. Caveolin, Flotillin or NOS) are not really “extracellular proteins”

I)

When talking about harmful “ligands/interactors” in neurodegenerative diseases – what about recent studies on PrPC as a receptor for (extracellular) alpha-synuclein or tau (although these are originally cytosolic proteins)?

Minor comments:

  1. When referencing other studies, “in vitro” or “in vivo” or “animals” are rather general terms. Prospective readers will likely profit from stating the exact experimental system.
  2. It would surely by more reader-friendly to change the order of the Alzheimer-related proteins (e.g. starting with APP – BACE – A-beta (maybe other toxic conformers such as tau or alpha-synuclein, too (see above)– mGluRs / LRP), maybe even combining this aspect to one chapter. This would create a better flow (and avoid redundancies such as the repeated introduction of the abbreviation “APP”).
  3. Why are kainate receptors and PSD-95 handled together in one chapter? Any reason for this?
  4. What about PrP and certain tetraspanins as increasingly important membrane regulators?
  5. Do TNAP and PrP really interact or is this all rather via laminin?
  6. There were some very nice and comprehensive reviews from Linden (on PrP interactors/functions) and Beland/Roucou (with a particular focus on ligands binding the N-terminal tail) some years ago – authors may consider referencing these reviews, too.
  7. Authors may also consider mentioning the increasingly anticipated role of interactions of physiologically produced/released PrP fragments or of PrP on extracellular vesicles with other proteins.

Author Response

Reviewer 2:

  1. The whole manuscript is filled with errors regarding grammatical issues, orthography, syntax and punctuation. Unfortunately, this makes it very difficult and tedious for readers to follow the overall flow of the text. For instance, many sentences are either lacking verbs or have it in the wrong place or form (e.g. tense, or plural vs singular); the article “the” is often used unnecessarily while missing elsewhere; “N-terminal” or “C-terminal” are adjectives, not nouns. Therefore, the text should undergo a careful and thorough proofreading by native English speakers. Also, several formatting errors can be found throughout the manuscript.

We thank the Reviewer 2 for this comment and apologize for the grammatical issues and errors. In addition to getting help from one our native English speaker colleagues, we also carefully checked the manuscript to correct those errors.

  1. For being a review on this topic, a more critical appraisal (maybe even deserving a separate chapter) would be highly appreciated: Can really all described “interactions” be considered “true” or -even more importantly- “physiologically relevant”? What about potential artefacts? A pull-down (e.g. from brain homogenates) may reveal several false “interactions”. What about technical limitations? PrP is known to be a “sticky” protein (note the use of detergents [e.g. 0.03% CHAPS] in recent ELISA studies of CSF samples to reduce PrP binding to plastic surfaces [group of Vallabh/Minikel]), so much “binding” to other proteins might be seen in lysates/homogenates without any in vivo relevance. Also, some parts/referenced studies deal with the so-called “neurotoxic” prion peptide 106-126 – is this /really a good proxy/analog to PrPC? I wonder how the multitude of interactions can really be true (considering protein/membrane topology and other constraints), yet I did not have the impression that the authors are wondering about this, too. Readers would likely be thankful for some “guidance” to understand these issues. What is the authors` or the referenced studies` view/explanation on this? How can PrPC interact with cytosolic proteins? Or is this interaction only suggested for cytosolic PrP or the other altered topology versions (if so, this should be clarified better)? The respective chapter 2 is somewhat confusing y the way – a scheme would surely be helpful (see point C below). Also, the individual subchapters introducing the particular proteins are mostly filled by a row of sentences mentioning -partially loosely connected findings/ facts. A bit more of conceptual guidance would clearly improve understanding.

We agree with the Reviewer 2 about the interference of plastic surface with PrPc content of biological samples. Both physisorption and chemisorption phenomena may exist with regards to analytes and in particular with proteins including PrPc.

The reviewer’s point is a very valuable one which unfortunately being ignored by most of the Life Science researchers. We also acknowledge that most of analytical methods used for the protein-protein interaction research have limitations and pitfalls. We highlighted this part in the conclusion, section 7.(lines 962-969, highlighted in pink).

We also added more detail and a figure on the cytosolic PrPc (Figure2)

-Although available methods cannot predict protein-protein interaction with 100% accuracy, computational methods will scale down a range of possible interactions to a collection of most likely interactions. These interactions will form a basis for further laboratory experiments. If used together, the gene expression data and protein interaction data would increase the trust on protein-protein interactions and the corresponding protein-protein interaction network.

-Residues 106-126 of the PrPc is toxic to cells expressing PrPc, but not to PrPc knockout neurons. We also agree that the 20 amino acid toxic residue cannot represent the whole feature of PrPc.

  1. Figures/schemes: It is somewhat strange that the only figure in a comprehensive review on PrP interactors is dealing with PrP mutations. Authors should consider additional (and more relevant) pictures/schemes (e.g. to improve understanding of topologically altered PrP forms, or to illustrate some important (and well-accepted) interactions and their (patho)physiological outcome).

We agree with the Reviewer’s valuable comment and added 3 more figures to the manuscript.

  1. Chapter 5 (Mutants/Polymorphisms) is not really connected well to the major topic of “interactions” – authors should elaborate on this a bit further to better integrate this aspect.

-More explanation was added to this chapter to highlight the importance of polymorphisms and mutations on PRNP gene. PRNP gene mutations may be induce conformational change that can affect the flexibility of the PrPc protein and consequently it may affect protein-protein interaction. (line 899-927, highlighted in teal)

  1. Several statements are misleading, confusing or even wrong. For instance, not whole PrPc -only its N-terminal half- is intrinsically disordered (Abstract); “purified from” is often used although “expression” is meant; line 184ff: “GFAP was found to be expressed more in the brain of scrapie-infected when compared to wild animals…” – I guess that “wild-type” is meant here (yet still, the comparison likely was between infected and non-infected mice, right?); various spellings can sometimes be found (e.g. Pint1 vs Pint 1 vs pint-1); line 244ff: “The highest tissue concentration of … is the brain and detects in …”; it is not true that A-beta is not found in non-demented controls; etc.)

- We appreciate the Reviewer 2 for his/her valuable comment and points. we corrected those sentences: sentence “As a consequence of its intrinsically disordered nature of the N-terminal, PrPc interacts with a wide range of protein partners.”

- We corrected all “purified from” in the text.

- We corrected the sentence to “GFAP was found to be expressed more in the brain of scrapie-infected mice when compared to non-infected mice “

- We changed all Pint-1 spelling.

- We changed “The highest tissue concentration of 14-3-3 protein is the brain” to “The highest expression of 14-3-3 protein is in the brain “

- We replaced “Aβ oligomers interact with the N-terminal domain of PrPc (amino acid 23-27 and 95-110) in AD brains and were never found in non-demented controls.” with “Aβ oligomers interact with the N-terminal domain of PrPc (amino acid 23-27 and 95-110) in AD brains “

  1. The conclusion contains some remarkable sentences that are much too general and/or much too far-reaching/exaggerated: e.g. “PrPC is identified as the main controller of cellular signaling …” or “… have still some undisclosed crucial roles…” (how can something unknown or undiscovered already be important or relevant?)

We completely agree with the Reviewer 2 and corrected those sentences:

  • We changed and corrected “PrPc is identified as the main controller of cellular signaling, functioning as a receptor and scaffold for different ligands and activating a variety of signal transduction pathways” to “PrPc can act as a cell signaling molecule, functioning as a receptor and scaffold for different ligands and activating a variety of signal transduction pathways.”
  • We corrected “PrPc, an evolutionary conserved molecule among the species, appears to have still some undisclosed crucial roles waiting to be deciphered beyond prion diseases” to “PrPc, an evolutionary conserved molecule among the species, needs further studies because a clear description of PrPc function still remains unclear.”

  1. The tables are surely helpful. However, I suggest to include (i) full names & abbreviation, (ii) information on the suggested main/primary function of that particular protein (class) (e.g. Pint- 1 (also termed PRNPIP or ERI3 is described as a RNA exonuclease), (iii) how / by which method and in which system interaction was described.

  • We added the full names & abbreviation, the class of protein and method and in which system interaction was described.

  1. Assignment: some listed proteins (e.g. Caveolin, Flotillin or NOS) are not really “extracellular proteins”

We agree with the reviewer 2 and the only purpose of

  • For better understanding we classified all PrPc protein partners in two groups: 1. Intracellular partners, 2. Extracellular and plasma membrane partners of PrPc. Intracellular partners of PrPc are those proteins that are in the cytosol or intracellular compartments such as nucleus, ER and mitochondria. Second group of PrPc partners includes proteins that are binding to the inner leaflet of PM, intramembrane proteins and extracellular proteins. (line 443, highlighted in violet)
  • All these 3 proteins are binding to the inner side of plasma membrane.

Caveolins are integral membrane proteins that are the main structural components of caveola membrane invaginations.

Flotillins are tightly associated with the inner leaflet of the plasma membrane

NOS can be found in both cytosol and the plasma membrane and the majority of nNOS activity is on the inner side of plasma membrane.

  1. When talking about harmful “ligands/interactors” in neurodegenerative diseases – what about recent studies on PrPC as a receptor for (extracellular) alpha-synuclein or tau (although these are originally cytosolic proteins)?

In the revised manuscript we added a paragraph on alpha-synuclein and discussed more details. Section 4.10. (line 618-628, highlighted in red)

  1. When referencing other studies, “in vitro” or “in vivo” or “animals” are rather general terms. Prospective readers will likely profit from stating the exact experimental system.

In the revised manuscript, we mentioned the exact experimental system instead of in vitro, in vivo and animals in the text.

  1. It would surely by more reader-friendly to change the order of the Alzheimer-related proteins (e.g. starting with APP – BACE – A-beta (maybe other toxic conformers such as tau or alpha-synuclein, too (see above)– mGluRs / LRP), maybe even combining this aspect to one chapter. This would create a better flow (and avoid redundancies such as the repeated introduction of the abbreviation “APP”).

For the revised manuscript, we added a cartoon to summarize all these interactions with PrPc. (figure 3)

  1. Why are kainate receptors and PSD-95 handled together in one chapter? Any reason for this?

- PrPc, PSD-95 and GluR5-7 are making a protein complex at the postsynaptic level to modulate JNK3 pathway. PSD-95 links GluR6 to JNK activation. Both PSD-95 and GluR6/7 have interaction with PrPc in the PM. The neurotoxicity induced by kainite depends on JNK3 pathway. PSD-95 is the mediator between kainite receptors and JNK3 pathway

  1. What about PrP and certain tetraspanins as increasingly important membrane regulators?

Tetraspanins-7 is immune molecule that has interaction with PrPc. We added more details about this interaction in chapter 5, “PrPc in the immune system”. (line 879-893, highlighted in light gray)

  1. Do TNAP and PrP really interact or is this all rather via laminin?

Ermonval et al. reported (Ermonval M, 2009) the TNAP-PrPc interaction using co-immunoprecipitation and mass spectrometry methods. They also showed that inhibition of TNAP activity alters the phosphorylation state of the PrPc-binding protein laminin, suggesting that TNAP and PrPc could functionally interact together. Further investigation is required to confirm that TNAP functional interaction with PrPc occurs directly or indirectly through the intermediate of other proteins (line 798-808, highlighted in blue)

  1. There were some very nice and comprehensive reviews from Linden (on PrP interactors/functions) and Beland/Roucou (with a particular focus on ligands binding the N-terminal tail) some years ago – authors may consider referencing these reviews, too.

We thank the reviewer 2 for his/her suggestion and added those references to the manuscripts.

- Linden R, Martins VR, Prado MA, Cammarota M, Izquierdo I, Brentani RR. Physiology of the prion protein. Physiol Rev. 2008;88(2):673-728.

- Chiarini LB, Freitas AR, Zanata SM, Brentani RR, Martins VR, Linden R. Cellular prion protein transduces neuroprotective signals. Embo j. 2002;21(13):3317-26.

- Beland M, Roucou X. The prion protein unstructured N-terminal region is a broad-spectrum molecular sensor with diverse and contrasting potential functions. J Neurochem. 2012;120(6):853-68.

  1. Authors may also consider mentioning the increasingly anticipated role of interactions of physiologically produced/released PrP fragments or of PrP on extracellular vesicles with other proteins.

We appreciate the Reviewer’s constructive comment and added new sentences to the revised manuscript:

 Another location that PrPc can be found is extracellular vesicle such as exosomes, highlighting the role of PrPc in intercellular communication. Both PrPc and PrPsc can efficiently transport with extracellular vesicles. The transportation element muskelin in association with cytoplasmic dynein and KIF5C binds to PrPc to efficiently traffic PrPc vesicle. Notably, muskelin handles two-way transport of PrPc and promotes lysosomal degradation over release of exosomal PrPc. Muskelin affect neurodegenerative condition by increasing lysosomal clearance of PrPc, limiting their PM and/or exosomal presentation. The connection between neuronal intracellular lysosome targeting of PrPc and extracellular exosome trafficking is related to the pathogenesis of neurodegenerative condition. (line 59-66, highlighted in red)

Round 2

Reviewer 1 Report

The manuscript has been improved according to the reviewer's suggestions. 

Author Response

Response to the Guest Editor:

We are grateful to the guest editor for considering our revised review manuscript. We thank the reviewer 1, for her/his supporting comments. We also thank the reviewer 2 for his/her constructive comments. We have revised the manuscript as per the Reviewer’s suggestions including a complete English language and writing style revision by one of our university colleagues (Prof. Robert Laprairie). We trust that the changes and corrections in the revised review manuscript has made it suitable for publication in the IJMS.  Thank you.

Sincerely,

Changiz Taghibiglou, PhD

Reviewer 1:

The manuscript has been improved according to the reviewer's suggestions. 

We thank the Reviewer 1 for his/her supporting comments.  To further improve the manuscript, we requested Dr. Robert Laprairie (Assistant Professor at the University of Saskatchewan) to assist us with the language editing and writing style. We also acknowledged his assistance in the Acknowledgement section.

Reviewer 2 Report

A) The manuscript has improved by the addition of new figures and additional aspects. However, although authors state that the text has undergone proofreading by a native speaker, I am afraid that there are still many mistakes regarding typos, grammar and formatting (e.g. some double spaces, diverse line spacing, etc) – some (yet by far not all) examples are listed below. As a consequence, the text still requires very thorough and consequent proofreading and editing before consideration of acceptance.

---

B) Unfortunately I was unable to assess certain new/modified parts due to a non-reader-friendly format to highlight changes (e.g. black text on dark green or dark blue; orange text on pink; etc.).

---

C) Scientific inconsistencies/unclarity can still be found all over: e.g. line 53/54: PrP (or it`s anchor) does not “bind to the” lipid rafts; it is attached to the membrane and -together with many other GPI-anchored and other proteins as well as certain typically enriched lipids- localized in (or even forming) the membrane domains termed DRMs or lipid rafts. / line 58-65: The newly added Muskelin (M) aspect is interesting yet confusing: I understand that M is an intracellular adaptor protein (“transportation element” might not be the most proper term) and involved in intracellular transport processes (not extracellular vesicle trafficking). / line 68: I guess “the translated form of PrPc” means nothing else than just “PrPc”? / regarding the other topological forms (figure and text), authors should use established nomenclature and only include the “c” in bona fide (i.e. correctly ‘positioned’) PrPc – not in the other forms.

---

Here are some EXAMPLES regarding point A:

As mentioned earlier, “C-terminal” or N-terminal” are adjectives, not nouns (e.g. line 53 and many other instances). / “Co-IP” is a noun, not a verb (e.g. line 167). / “Pint-1” is still written in diverse ways (e.g. non-capital vs capital, lines 212/213). / “The” has been added without need in many instances. / missing verbs (e.g. line 897: “…and how pathogenic PRNP mutations (are) involved in prion diseases…”) / line 918: “polybasic” instead of “poly basic”.

Singular vs plural; e.g.: “Another subcellular organelles that PrPc can be found is extracellular vesicles such as exosomes, …” (line 57/58) / What is “a numerous group of…” (line 138). / “In the inherited form of prion disease,…” – well, there are different forms (line 900).

The abbreviation for Amyloid -beta only needs to be introduced on first mention – not several times in a row.

Author Response

Response to the Guest Editor:

We are grateful to the guest editor for considering our revised review manuscript. We thank the reviewer 1, for her/his supporting comments. We also thank the reviewer 2 for his/her constructive comments. We have revised the manuscript as per the Reviewer’s suggestions including a complete English language and writing style revision by one of our university colleagues (Prof. Robert Laprairie) . We trust that the changes and corrections in the revised review manuscript has made it suitable for publication in the IJMS.  Thank you.

Sincerely,

Changiz Taghibiglou, PhD

Reviewer 2:

English language and style

(x) Extensive editing of English language and style required 

We thank the Reviewer 2 for this comment and apologize for the grammatical issues and errors. To improve the manuscript, we requested Dr. Robert Laprairie (Assistant Professor at the University of Saskatchewan) to assist us with the language editing and writing style. We also acknowledged his assistance in the Acknowledgement section.

  1. A) The manuscript has improved by the addition of new figures and additional aspects. However, although authors state that the text has undergone proofreading by a native speaker, I am afraid that there are still many mistakes regarding typos, grammar and formatting (e.g. some double spaces, diverse line spacing, etc) – some (yet by far not all) examples are listed below. As a consequence, the text still requires very thorough and consequent proofreading and editing before consideration of acceptance.

Sorry for all those errors. We carefully checked the manuscript to correct those errors.

  1. B) Unfortunately I was unable to assess certain new/modified parts due to a non-reader-friendly format to highlight changes (e.g. black text on dark green or dark blue; orange text on pink; etc.).

Sorry for the inconvenience. We highlighted all the revisions with light gray.

  1. C) Scientific inconsistencies/unclarity can still be found all over: e.g. line 53/54: PrP (or it`s anchor) does not “bind to the” lipid rafts; it is attached to the membrane and -together with many other GPI-anchored and other proteins as well as certain typically enriched lipids- localized in (or even forming) the membrane domains termed DRMs or lipid rafts.

We rewrote and corrected this sentence in the revised manuscript.

line 58-65: The newly added Muskelin (M) aspect is interesting yet confusing: I understand that M is an intracellular adaptor protein (“transportation element” might not be the most proper term) and involved in intracellular transport processes (not extracellular vesicle trafficking).

The transportation element muskelin in association with cytoplasmic dynein and KIF5C, binds to PrPc to efficiently traffic PrPc vesicles. Muskelin involves in intracellular transport. Heisler et al. reported that in the absence of muskelin, PrPc is not targeted to the lysosome but instead it is recycled to the PM and sorted for extracellular trafficking via exosomal carriers. We added more explanations to the revised manuscript to clarify the role of Muskelin in the PrPc vesicles trafficking.

line 68: I guess “the translated form of PrPc” means nothing else than just “PrPc”? / regarding the other topological forms (figure and text), authors should use established nomenclature and only include the “c” in bona fide (i.e. correctly ‘positioned’) PrPc – not in the other forms.

We agree with the Reviewer’s comment and we edited the figure to show the correctly positioned PrPc.

Here are some EXAMPLES regarding point A:

As mentioned earlier, “C-terminal” or N-terminal” are adjectives, not nouns (e.g. line 53 and many other instances). / “Co-IP” is a noun, not a verb (e.g. line 167). / “Pint-1” is still written in diverse ways (e.g. non-capital vs capital, lines 212/213). / “The” has been added without need in many instances. / missing verbs (e.g. line 897: “…and how pathogenic PRNP mutations (are) involved in prion diseases…”) / line 918: “polybasic” instead of “poly basic”.

Singular vs plural; e.g.: “Another subcellular organelles that PrPc can be found is extracellular vesicles such as exosomes, …” (line 57/58) / What is “a numerous group of…” (line 138). / “In the inherited form of prion disease,…” – well, there are different forms (line 900).

The abbreviation for Amyloid -beta only needs to be introduced on first mention – not several times in a row.

We appreciate the Reviewer 2 comments and corrected those mistakes in the revised manuscript.

Round 3

Reviewer 2 Report

Reviewer evaluation of submission ijms-901231-v3

I appreciate involvement of a native speaker in the proofreading process. However, I just briefly skimmed over the text and, unfortunately, still found it spiked with typos, confusing expression, as well as grammatical and formatting errors. Below I provide yet another list of a few examples (and when stating “examples” I really mean that it is not a complete list of all those mistakes – thus, I do not expect a point-by-point response on those issues but rather a significantly improved version as a whole in the best interest of readability and understanding of prospective readers). Frankly speaking, I do not consider this being part of the reviewers` task and therefore will not be available for another round of corrections.

  • The improper use (and also “non-use”) of the article “the” is striking throughout the text.

  • format of text in tables (left-aligned vs centered)

  • line 57: “…found is extracellular vesicles…” (replace “is” by “in”)

  • I do not understand the sentence in lines 454/455

  • line 553: “α -synuclein” – α-synuclein

  • line 589: “… channels tha tare widespread…” - that are

  • line 601: “…Cu 2+and…” - Cu2+ and

  • numbers until twelve should be spelled (no number); this would improve readability (for instance in line 644: “…, 2 140 kDa and 180 kDa isoforms…” - two 140 kDa…)

  • line 784: “group” – groups

  • line 867/868: “.Tetraspanin-7 widely…” - .(space) Tetraspanin-7 is widely…

  • line 883: “protease-resistant-β sheet rich” – protease-resistant β-sheet(-)rich

  • line 874: “prior disease” – prion disease

Also, some (again only EXAMPLES) logical/scientific unclarities remain:

  • Abstract: “prion neurodegenerative diseases” - why not “neurodegenerative prion diseases”?

  • Topological forms: I guess there may be a misunderstanding regarding my previous comments: The “c” in PrPc (not C-terminus) should be deleted when referring to the other topological forms (e.g. ctmPrP instead of ctmPrPc). Also the text is confusing here: ctmPrP and ntmPrP are not “cytosolic forms” – they are transmembrane forms instead (as correctly drawn in the figure). By the way: I just realized that the bona fide “cytosolic” form is not even shown in the figure (although some interactors and pathophysiological roles have been suggested in the literature).

  • I wonder why tetraspanins are per se discussed as “immune molecules” and not as what they are in the first place (and in more general terms): i.e., a family of membrane (domain) organizing (“tetraspanin web”) and trafficking-regulating proteins (of course with roles in immune (as well as many other) processes)… Also, the text is somewhat confusing as it starts with “singular” (“The other molecule is tetraspanin”) to later introduce the whole family (and hence multitude of different “members”).

Author Response

Response to the Reviewer 2:

We thank the Reviewer 2 for his/her time and constructive comments. We went through the manuscript and corrected all the points raised by the Reviewer 2.

Topological forms: I guess there may be a misunderstanding regarding my previous comments: The “c” in PrPc (not C-terminus) should be deleted when referring to the other topological forms (e.g. ctmPrP instead of ctmPrPc). Also the text is confusing here: ctmPrP and ntmPrP are not “cytosolic forms” – they are transmembrane forms instead (as correctly drawn in the figure). By the way: I just realized that the bona fide “cytosolic” form is not even shown in the figure (although some interactors and pathophysiological roles have been suggested in the literature).

We agree with the Reviewer’s comment and revised the figure to show the correctly positioned PrPc in the ER membrane. In section “2. Transmembrane forms of PrPc”, we intended to discuss about the topological form of PrPc in the PM so we just showed tbe NtmPrPc, CtmPrPc and correct position of PrPc in the ER membrane.

I wonder why tetraspanins are per se discussed as “immune molecules” and not as what they are in the first place (and in more general terms): i.e., a family of membrane (domain) organizing (“tetraspanin web”) and trafficking-regulating proteins (of course with roles in immune (as well as many other) processes)… Also, the text is somewhat confusing as it starts with “singular” (“The other molecule is tetraspanin”) to later introduce the whole family (and hence multitude of different “members”)

According to previous studies, tetraspanin-7 was introduced as an immune molecule (Guo et all, 2007). The immune system contributes to pathogenesis partly by amplifying PrPc in lymphoid compartments, so facilitating efficient neuro-invasion. Moreover, incubation of T cells with anti-PrPc monoclonal antibodies results in internalization of a large proportion of surface PrPc into Limp-2 positive endosomes. So, further investigations on the PrPc/tetraspanin-7 complex will help to understand how immune cells respond to PrPc.